# Serine palmitoyltransferase assembles at ER–mitochondria contact sites

Mari J Aaltonen[1,2] , Irina Alecu[3,4], Tim König[1] , Steffany AL Bennett[3,4], Eric A Shoubridge[1,2]

The accumulation of sphingolipid species in the cell contributes to the development of obesity and neurological disease. However, the subcellular localization of sphingolipid-synthesizing enzymes is unclear, limiting the understanding of where and how these lipids accumulate inside the cell and why they are toxic. Here, we show that SPTLC2, a subunit of the serine palmitoyltransferase (SPT) complex, catalyzing the first step in de novo sphingolipid synthesis, localizes dually to the ER and the outer mitochondrial membrane. We demonstrate that mitochondrial SPTLC2 interacts and forms a complex in trans with the ER-localized SPT subunit SPTLC1. Loss of SPTLC2 prevents the synthesis of mitochondrial sphingolipids and protects from palmitate-induced mitochondrial toxicity, a process dependent on mitochondrial ceramides. Our results reveal the in trans assembly of an enzymatic complex at an organellar membrane contact site, providing novel insight into the localization of sphingolipid synthesis and the composition and function of ER–mitochondria contact sites.

## Introduction

The lipid composition of cellular membranes is determined by the specific localization of proteins mediating lipid synthesis and transport. Most lipids are synthesized in the ER and need to be transported to other organelles via vesicles or by lipid transfer proteins at membrane contact sites. Contact sites are essential for pathways and functions requiring organellar communication. For example, proteins from the mitochondrial outer membrane (OM) and mitochondria-associated ER membrane (MAM) interact to allow exchange of molecules, such as phospholipids, at ER–mitochondria contact sites (Scorrano et al, 2019; Prinz et al, 2020).

Serine palmitoyltransferase (SPT) catalyzes the first and rate-limiting step of de novo sphingolipid synthesis by producing 3-keto-sphinganine from L-serine and palmitoyl-CoA (Fig 1A) (Merrill & Wang, 1986). Via a multi-enzymatic cascade, this sphingoid base is converted to ceramide through the addition of an N-acyl chain, and

further converted to a multitude of sphingolipid species, including sphingosine, glycosphingolipids, and gangliosides (Pruett et al, 2008). The SPT core complex is formed by two subunits: SPTLC1 and the cofactor pyridoxal-phosphate containing SPTLC2, or its homolog SPTLC3, and presence of both subunits is essential for enzyme activity (Lone et al, 2020). SPT formed by ubiquitously expressed SPTLC1 and SPTLC2 preferably synthesizes the palmitoyl-containing sphingoid base (Lone et al, 2020), which is the most abundant sphingoid base in mammalian cells (Pruett et al, 2008), whereas SPTLC3 expression is restricted to specific tissues and leads to the production of short-, long-, and branched-chain sphingolipid species (Hornemann et al, 2006, 2009; Lone et al, 2020).

Mutations in *SPTLC1* and *SPTLC2* cause hereditary sensory and autonomic neuropathy type 1, a rare neurological disease (Dawkins et al, 2001; Rotthier et al, 2010) and macular telangiectasia type 2 (Gantner et al, 2019). Pathogenic variants in SPT have been shown to prefer L-alanine over L-serine, resulting in the production of harmful lipotoxic deoxy-sphingolipids that cannot be converted to complex sphingolipids or degraded (Gable et al, 2010; Penno et al, 2010; Bode et al, 2016). Recently, SPT-dependent deoxy-sphingolipid synthesis was shown to inhibit tumor growth in vivo and in vitro (Muthusamy et al, 2020), and excess sphingolipid synthesis by pathogenic SPTLC1 variants was reported to cause childhood amyotrophic lateral sclerosis (Mohassel et al, 2021).

Not only the synthesis of harmful lipids, but also the subcellular compartment in which they accumulate, can influence how cells cope with the increased lipid load (reviewed in Turpin-Nolan and Bruning [2020]). Ceramide synthases (CERS) and their enzymatic activity have been reported to be present in mitochondria (Ullman & Radin, 1972; Shimeno et al, 1995, 1998; Bionda et al, 2004; Yu et al, 2007; Novgorodov et al, 2011; Oleinik et al, 2019), and exogenously added alkyne-labeled sphinganine and deoxy-sphinganine have been shown to localize to mitochondria where deoxy-sphingolipids specifically cause mitochondrial dysfunction leading to neuronal cell defects in vitro (Alecu et al, 2017; Wilson et al, 2018). Dysfunctional sphingolipid metabolism is tightly linked to metabolic disease (reviewed in Meikle and Summers [2017]). Ceramide accumulation, particularly in mitochondria and MAMs, contributes to the development of obesity and insulin resistance in mice, which is

---

[1]Montreal Neurological Institute, McGill University, Montreal, Canada   [2]Department of Human Genetics, McGill University, Montreal, Canada   [3]Department of Biochemistry, Microbiology, and Immunology, Faculty of Medicine, University of Ottawa, Ottawa, Canada   [4]Ottawa Institute of Systems Biology, Ottawa, Canada

Correspondence: eric.shoubridge@mcgill.ca

 

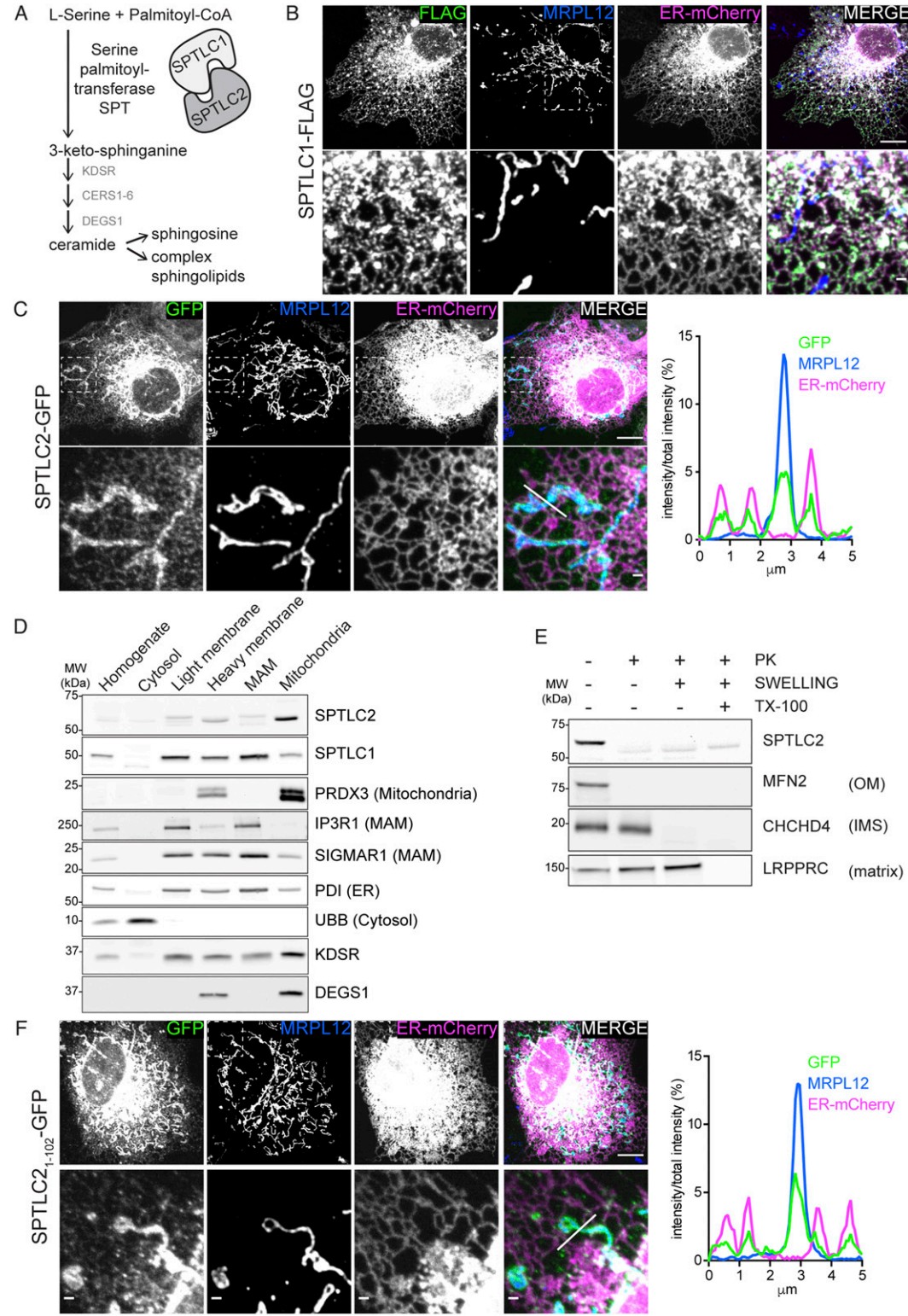

**Figure 1. SPTLC2 localizes to the ER and the mitochondrial outer membrane.**
**(A)** Schematic of the SPT complex formed by SPTLC1 and SPTLC2 and its role in de novo sphingolipid synthesis. **(B)** Confocal microscopy images of SPTLC1 localization to the ER. SPTLC1-FLAG was transiently expressed in COS-7 cells and visualized using an anti-FLAG antibody. ER-mCherry serves as an ER marker and endogenous MRPL12 serves as a mitochondrial marker. Scale bar 10 or 1 μm (zoom). **(C)** Confocal microscopy images and line scan of fluorescence intensities demonstrating SPTLC2 localization to mitochondria and ER. SPTLC2-GFP was transiently expressed in COS-7 cells and visualized by GFP signal enhanced with anti-GFP-488 antibody. ER-mCherry serves as ER marker and MRPL12 serves as mitochondrial marker. Scale bar 10 or 1 μm (zoom). Line scan (from left to right) of fluorescence intensities normalized to total intensity of

prevented by the ablation of the $C_{16:0}$ ceramide producing enzyme ceramide synthase 6 (CerS6) (Raichur et al, 2014; Turpin et al, 2014; Hammerschmidt et al, 2019). Accumulation of CerS6-derived $C_{16:0}$ ceramides causes mitochondrial fragmentation in vivo and in vitro, and CerS6-produced sphingolipids specifically interact with mitochondrial proteins (Hammerschmidt et al, 2019).

Despite growing evidence of the presence, synthesis, and importance of ceramides in mitochondria, it remains unknown how ceramide precursors reach mitochondria. Here we describe the presence of SPT at ER–mitochondria contact sites where mitochondrial OM-localized SPTLC2 interacts in trans with ER-localized SPTLC1 controlling palmitate-induced mitochondrial fragmentation.

# Results and Discussion

### SPTLC2 localizes to mitochondria

To define the subcellular localization of SPTLC1 and SPTLC2, we analyzed C-terminally tagged SPTLC1 and SPTLC2, transiently expressed in COS-7 cells, by confocal microscopy. SPTLC1-FLAG showed complete co-localization with the ER-marker Sec61b-mCherry (Fig 1B), as previously reported (Yasuda et al, 2003). These results were recapitulated by anti-SPTLC1 staining at endogenous protein levels (Fig S1A). Unexpectedly, SPTLC2-GFP co-localized not only with the ER-marker but also with the mitochondrial marker MRPL12 (Fig 1C). Consistent with these results, subcellular fractionation of mouse liver showed that endogenous SPTLC1 and SPTLC2 were present in the microsomal light membrane fraction, and in the heavy membrane fraction containing mitochondria and MAMs (Fig 1D). Gradient-purification of the heavy membranes into MAMs and mitochondria revealed that SPTLC2 was enriched in mitochondria, whereas SPTLC1 was enriched in MAMs (Fig 1D). SPT was long assumed to reside solely on the ER, based on the enrichment of SPT activity in the microsome fraction (Williams et al, 1984). However, early activity assays and fractionation protocols (De Duve et al, 1955) did not consider the presence of ER–mitochondria contact sites, as MAMs were only discovered as a hotspot for phospholipid synthesis in 1990 (Vance, 1990). SPTLC2 is included in the Mitocarta3.0 mitochondrial protein database, supporting its mitochondrial localization (Rath et al, 2021).

The presence of SPTLC2 in mitochondria and previous evidence of CERS localization to mitochondria, raises the question of whether the two remaining enzymes in the de novo sphingolipid synthesis pathway, namely 3-keto-sphinganine reductase KDSR (which reduces SPT-formed 3-keto-sphinganine into sphinganine) and dihydroceramide desaturase DEGS (producing ceramide by insertion of a double bond in CERS-formed dihydroceramide) are also present on mitochondria. KDSR has so far only been reported on the ER membrane (Kihara & Igarashi, 2004), whereas DEGS1 has been shown to be dually localized to the ER and mitochondria (Beauchamp et al, 2009) and DEGS1 KO leads to reduced mitochondrial ceramide levels (Siddique et al, 2013). Subcellular fractionation revealed dual localization of KDSR to light membranes and mitochondria, and enrichment of DEGS1 in gradient purified mitochondria (Fig 1D). DEGS1 localization to mitochondria could also be observed in A431 cells by immunofluorescence with an anti-DEGS1 antibody (Fig S1B).

To interrogate the topology of SPTLC2 in mitochondria, we performed a protease protection assay on mitochondria-enriched heavy membrane fractions isolated from Flp-In T-Rex 293 cells. Addition of proteinase K leads to the degradation of surface-exposed proteins and the disappearance of OM protein signals on immunoblot analysis, whereas the intermembrane space (IMS) and matrix proteins are only degraded upon hypotonic swelling causing rupturing of the OM or Triton X-100 solubilization, respectively. Similar to the OM protein MFN2, SPTLC2 was accessible to the protease in intact mitochondria, whereas the IMS protein CHCHD4 and matrix protein MRPL44 were protected (Fig 1E), suggesting that SPTLC2 is an OM protein exposed to the cytosol. SPTLC1 and SPTLC2 share sequence similarity (25% identity), particularly within their conserved C-terminal aminotransferase domains which constitute most of the proteins; however, their N termini differ in amino acid sequence and length (Fig 1C). SPTLC2 has an extended N terminus resulting in a protein with larger molecular weight (63 kD, 562 amino acids) than SPTLC1 (53 kD, 473 amino acids). We reasoned that the N terminus of SPTLC2 could be responsible for targeting the protein to mitochondria. To test this, the first 102 amino acids of SPTLC2 were fused in front of GFP and transiently expressed in COS-7 cells. Truncated $SPTLC2_{1-102}$ was sufficient to target GFP to both mitochondria and ER (Fig 1F), showing that the dual localization of SPTLC2 is determined by the first 102 N-terminal amino acids. Removal of the N-terminal 102 residues ($SPTLC2_{103-562}$-GFP) led to a uniform cytosolic distribution (Fig S1D). Both SPTLC1 and SPTLC2 have one predicted transmembrane helix in the N terminus (Fig S1C), and SPTLC1 has been experimentally shown to be a membrane protein (Yasuda et al, 2003). The subcellular fractionation of mouse liver (Fig 1D) suggested that SPTLC2 is also associated with membranes. Upon alkaline sodium carbonate extraction of heavy membrane fractions isolated from 293 Flp-In T-Rex cells, SPTLC1, and SPTLC2 were found predominantly in the pellet fraction, similarly to the one-transmembrane-containing OM proteins MFN2 and TOMM20, whereas membrane-associated SDHA and soluble GRSF1 were found in the supernatant (Fig S1E). In summary, two subunits of the SPT-complex, SPTLC1 and SPTLC2 show different subcellular localizations: SPTLC1 localizes to the ER membrane, whereas SPTLC2 is present both on the ER membrane and mitochondrial OM.

---

the channel. **(D)** Subcellular localization of SPTLC2 and SPTLC1 in vivo. Mouse liver was fractionated and the fractions were analyzed by SDS–PAGE and immunoblotting. **(E)** Submitochondrial localization of SPTLC2. Protease protection assay on crude mitochondria isolated from Flp-In T-REx 293 cells subjected to hypotonic swelling or solubilization with Triton X-100 (TX-100). Where indicated, samples were treated with proteinase K (PK). **(F)** Confocal microscopy images and line scan of fluorescence intensities demonstrating that the N-terminus of SPTLC2 targets mitochondria and ER. $SPTLC2_{1-102}$-GFP was transiently expressed in COS-7 cells and visualized by GFP signal enhanced with anti-GFP-488 antibody. ER-mCherry serves as an ER marker and MRPL12 serves as a mitochondrial marker. Scale bar 10 or 1 $\mu$m (zoom). Line scan (from left to right) of fluorescence intensities normalized to total intensity of the channel.
Source data are available for this figure.

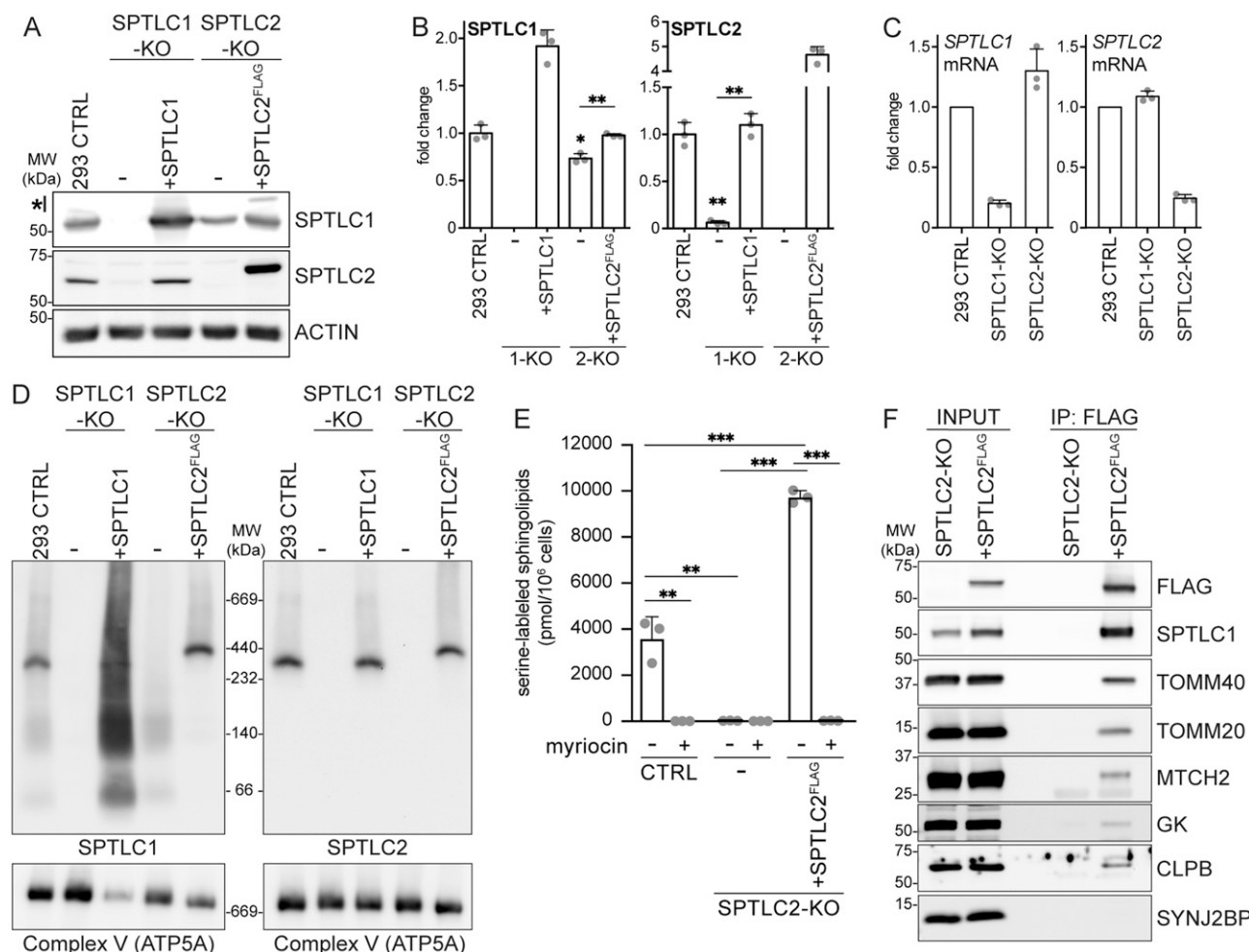

**Figure 2. Analysis of SPTLC1-KO and SPTLC2-KO cells.**
**(A)** SPTLC1 and SPTLC2 protein levels in SPTLC1-KO and SPTLC2-KO cells. Whole cell lysates from Flp-In T-REx 293 cells were analyzed by SDS–PAGE and immunoblotting. *, carryover signal from SPTLC2 decoration. Actin was used as a loading control. **(A, B)** Quantification of SPTLC1 (left panel) and SPTLC2 (right panel) protein levels in SPTLC1-KO (1-KO) and SPTLC2-KO (2-KO) from (A). SPTLC1 and SPTLC2 were normalized to Actin signal. Mean ± SD n = 3 independent replicates, unpaired two-sided Welch's *t* test, *P < 0.05, **P < 0.01. **(C)** qRT-PCR analysis of *SPTLC1* (left panel) and *SPTLC2* (right panel) mRNA levels relative to control. Mean ± SD n = 3 independent replicates. **(D)** Characterization of the SPT complex. Heavy membrane fractions from Flp-In T-REx 293 cells, SPTLC-KOs and rescues were analyzed by blue native PAGE. Samples were analyzed in duplicate on the same gel and immunoblotted with anti-SPTLC1 (left) and anti-SPTLC2 antibodies (right). ATP5A antibody was used as a loading control. **(E)** Quantification of newly synthesized sphingolipids in cells. A431 control, SPTLC2-KO, and SPTLC2-KO cells constitutively expressing SPTLC2^FLAG were grown in the presence of (2,3,3-D3, ^15N)-L-serine for 24 h. Lipids were extracted from cell pellets and stable isotope-labeled sphingolipids were analyzed by liquid chromatography–mass spectrometry. SPT-inhibitor myriocin was added for the duration of the labeling when indicated. Mean ± SD n = 3 independent replicates, unpaired two-sided *t* tests corrected for multiple comparisons, **P < 0.01, ***P < 0.001. **(F)** Immunoblot analysis of proteins co-purified with SPTLC2^FLAG. Heavy membrane fractions from Flp-In T-REx 293 SPTLC2-KO and SPTLC2-KO cells expressing SPTLC2^FLAG were solubilized by digitonin, subjected to FLAG-immunoprecipitation, and input (5%) and eluate (IP:FLAG, 40%) fractions were analyzed by SDS–PAGE and immunoblotting. SYNJ2BP was used as a negative control.
Source data are available for this figure.

## Characterization of SPTLC2-KO cells and protein interactions

The localization of SPTLC1 and SPTLC2 on opposing membranes suggested the possibility that SPTLC1 and SPTLC2 could interact in trans across organelles to form a functional SPT-complex, in addition to their interaction in cis on the ER membrane. To carefully characterize the interaction of mitochondrial SPTLC2 with SPTLC1 in the ER, we first generated CRISPR-Cas9–mediated KO of SPTLC1 and SPTLC2 in Flp-In T-REx 293 cells, which do not express *SPTLC3* (Hornemann et al, 2006). The KO of SPTLC1 resulted in an almost complete loss of SPTLC2 (6% of control) (Fig 2A and B), whereas

*SPTLC2* mRNA expression was not changed (Fig 2C), suggesting that SPTLC1 is required for SPTLC2 protein stability, as noted in previous reports (Yasuda et al, 2003; Lone et al, 2020). The SPTLC2 level was rescued by the re-expression of SPTLC1 (Fig 2A and B). The KO of SPTLC2 did not affect the expression of *SPTLC1* mRNA (Fig 2C) but lead to a slightly reduced level of SPTLC1 protein (73% of control), suggesting that SPTLC2 also affects SPTLC1 stability. The SPTLC1 level was rescued by the re-expression of SPTLC2^FLAG (Fig 2A and B). The FLAG-tagged SPTLC2 behaved similarly to SPTLC2-GFP and the endogenous SPTLC2 in that it fractionated with purified mitochondria and light membranes (Fig S2A), was integrally inserted

into the mitochondrial OM in Flp-In T-REx 293 cells (Fig S2B and C), and showed ER and mitochondrial localization upon expression in COS-7 cells (Fig S3A).

To study the interaction of SPTLC1 and SPTLC2 in the SPT complex, we set out to detect the SPT complex on a native gel where protein–protein interactions are preserved. SPT appeared as oligomeric complexes in size-exclusion chromatography and semi-denaturing blue-native PAGE (Hornemann et al, 2007). In our hands, the separation of digitonin-solubilized heavy membrane fractions on a native gel, followed by immunoblotting with SPTLC1 and SPTLC2 antibodies revealed an approximately 350~400-kD SPT complex (Fig 2D). Both SPTLC1-KO and SPTLC2-KO cells displayed a complete loss of this complex, which was rescued upon re-expression of SPTLC1 and SPTLC2$^{FLAG}$, respectively, validating the interaction of SPTLC1 and SPTLC2 in this complex. In SPTLC2-KO cells, SPTLC1 appeared in diffuse lower molecular weight complexes likely representing monomeric and dimeric SPTLC1, as these complexes were also increased upon SPTLC1 overexpression in rescued SPTLC1-KO cells (Fig 2D).

For further analysis of SPTLC2, a KO of SPTLC2 was generated in human A431 epidermoid carcinoma cells. A rescue cell line was generated by stable integration of constitutively expressed SPTLC2$^{FLAG}$ (Fig S2D), at a protein level similar to control cells. To determine the SPT-activity in the generated cells, we measured the incorporation of a stable isotope-labeled serine into newly synthesized sphingolipids by liquid chromatography–mass spectrometry. A431 cells showed prominent incorporation of the stable isotope-labeled serine into sphingolipids, with very long-chain $C_{24}$-N-acylated ceramides and hexosylceramides as the most abundant lipid species identified (Fig 2E, the identified lipid species are detailed in Table S1). KO of SPTLC2 in A431 cells abolished SPT activity, which could be rescued by re-expression of SPTLC2$^{FLAG}$ (Fig 2E). Lipid synthesis was prevented in the presence of SPT-inhibitor myriocin (Fig 2E).

To identify interacting proteins of SPTLC2, heavy-membrane fractions were isolated from SPTLC2-KO cells expressing SPTLC2$^{FLAG}$ and SPTLC2-KO cells as control. SPTLC2$^{FLAG}$ was purified by FLAG-immunoprecipitation, eluate fractions were analyzed by mass spectrometry and SPTLC2$^{FLAG}$-specific proteins were ranked by their Mascot scores. As expected, the highest-scoring interactor of SPTLC2$^{FLAG}$ was SPTLC1 (Table S2). Also, the previously described SPTLC1 interactor, ER protein ORMDL (Breslow et al, 2010) and its homologs ORMDL1 and ORMDL2 were co-immunoprecipitated with SPTLC2$^{FLAG}$. Other detected ER proteins included ESYT1, EMD, AMFR, and CANX (Table S2). Strikingly, several mitochondrial proteins also co-purified with SPTLC2$^{FLAG}$, including MFF which was previously reported to interact with mitochondrial sphingolipids (Hammerschmidt et al, 2019). Reflecting the OM localization of SPTLC2, most of the co-immunoprecipitated mitochondrial proteins were known OM proteins, such as TOMM40, TOMM20, MTCH2, and GK, and also the IMS protein CLPB. These interactions were confirmed by immunoblotting, where the OM protein SYNJ2BP served as a negative control (Fig 2F).

## Mitochondrial SPTLC2 interacts in trans with SPTLC1 at ER–mitochondria contact sites

After establishing the assays to look at the interaction of SPTLC2 with SPTLC1, we set out to test whether OM-localized SPTLC2 could interact with the ER-localized SPTLC1. First, constructs expressing

OM- and ER-specific SPTLC2 were engineered by replacing the N-terminal targeting peptide (amino acids 1–102) of SPTLC2 with OM- or ER-targeting peptides, and adding a 3xFLAG peptide to the C-terminus (Fig 3A). In detail, to localize SPTLC2 exclusively to the ER, the N terminus of SPTLC1 (amino acids 1–49 including the transmembrane domain) was inserted in front of the C-terminus of SPTLC2 (amino acids 103–562). To achieve mitochondrial OM localization, the N terminus of the OM protein mAKAP1 (amino acids 1–31 including the transmembrane domain) together with a short positively charged linker peptide, was fused in front of the C ter-minus of SPTLC2. Changing the N terminus of SPTLC2 is not expected to interfere with its interaction with SPTLC1 because structural analysis revealed that the interaction is formed by the C-terminal aminotransferase domains (Yard et al, 2007; Wang et al, 2021).

Transient expression in COS-7 cells confirmed that ER-SPTLC2$^{FLAG}$ indeed localized exclusively to the ER and OM-SPTLC2$^{FLAG}$ to mi-tochondria, whereas WT-SPTLC2$^{FLAG}$ was dually localized (Fig S3A). ER- and OM-specific SPTLC2$^{FLAG}$ constructs were then stably inte-grated into the SPTLC2-KO Flp-In T-REx 293 cells under a tetracy-cline inducible promoter. To investigate their interaction with ER-localized SPTLC1, mitochondria-enriched heavy membrane fractions were isolated from SPTLC2-KO and SPTLC2-KO cells expressing WT-SPTLC2$^{FLAG}$, ER-SPTLC2$^{FLAG}$, and OM-SPTLC2$^{FLAG}$. After FLAG im-munoprecipitation, eluate fractions were analyzed by SDS–PAGE and immunoblotting. OM-SPTLC2$^{FLAG}$ retained the interaction with the OM proteins TOMM40 and TOMM20, whereas ER-SPTLC2$^{FLAG}$ lost these interactions. SPTLC1 was co-immunoprecipitated with both ER-SPTLC2$^{FLAG}$ and OM-SPTLC2$^{FLAG}$ (Fig 3B), illustrating that SPTLC2 can interact with SPTLC1 on cis and trans membranes.

Interaction of OM-targeted SPTLC2 with ER-localized SPTLC1 suggests that SPT complex assembles at ER–mitochondria contact sites. Transiently overexpressed SPTLC2$^{GFP/FLAG}$ uniformly distrib-uted on mitochondria (Figs 1C and S3A), but considering that SPTLC2 is stable only in a complex with ER-located SPTLC1 (Fig 2A–D), we hypothesized that mitochondrial SPTLC2, when expressed at en-dogenous level, should be stable only at contact sites with the ER. To demonstrate localization of SPTLC2 to ER–mitochondria contact sites, we perfomed super-resolution microscopy in SPTLC2-KO cells re-expressing stably integrated SPTLC2$^{FLAG}$ at a level similar to that of the endogenous protein (Fig S3B) in U2OS cells where tubular mito-chondria and ER can be nicely visualized. Super-resolution microscopy revealed puncta of SPTLC2$^{FLAG}$ localized to the ER and to mitochondria at ER contact sites (Fig 3C). Similar SPTLC2$^{FLAG}$ distribution by super-resolution microscopy could also be seen in A431 SPTLC2-KO cells re-expressing SPTLC2$^{FLAG}$ at near endogenous levels (Fig S3C and D).

To observe the SPT complex at ER–mitochondria contact sites on a native gel, we analyzed mouse liver light and heavy membrane fractions (obtained in Fig 1D). The SPT complex could be observed in mouse liver light membrane fraction, where both SPTLC1 and SPTLC2 are on the ER (Fig 3D). In the heavy membrane fraction where SPTLC1 is on the MAMs and SPTLC2 in mitochondria (Fig 1D), SPT assembled into a similar molecular weight complex, suggesting that endogenous SPT complex assembles across organelles (Fig 3D). In gradient-purified mitochondria where MAMs have been stripped off by gradient-purification, the complex is almost com-pletely lost and SPTLC2 is present mainly as a monomer because of the absence of MAM-localized SPTLC1 in this fraction (Fig 3D).

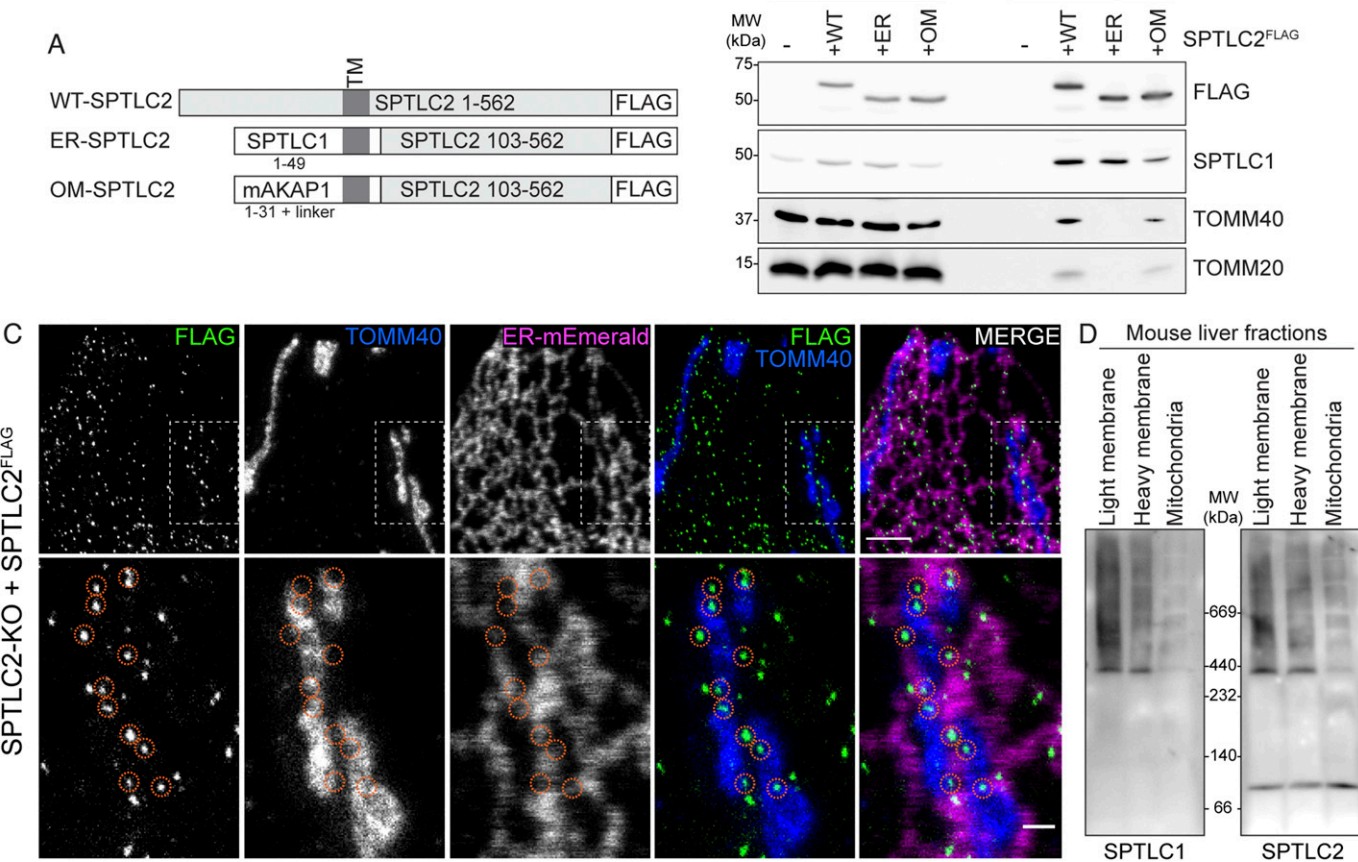

**Figure 3. Mitochondrial SPTLC2 interacts with SPTLC1 in the ER.**
**(A)** Schematic of WT-, ER-, and OM-SPTLC2[FLAG] constructs. **(B)** Immunoblot analysis of SPTLC1 co-purification with WT-, ER-, and OM-SPTLC2[FLAG]. Heavy membrane fractions from Flp-In T-REx 293 SPTLC2-KO and SPTLC2-KO cells expressing WT-, ER-, and OM-SPTLC2[FLAG] were solubilized by digitonin, subjected to FLAG-immunoprecipitation, and input (5%) and eluate (25%) fractions were analyzed by SDS–PAGE and immunoblotting. **(C)** Super-resolution microscopy images of SPTLC2[FLAG] localization. SPTLC2[FLAG] was stably integrated into SPTLC2-KO U2OS cells and visualized by FLAG signal. ER-Emerald serves as ER marker and TOMM40 serves as mitochondrial marker. Orange circles highlight SPTLC2[FLAG] puncta at ER–mitochondria contact sites. Scale bar 2 or 0.5 µm (zoom). **(D)** Characterization of the SPT complex in mouse liver fractions. Light and heavy membrane fractions and purified mitochondria from mouse liver were analyzed by blue native PAGE. Samples were analyzed in duplicate on the same gel and immunoblotted with anti-SPTLC1 (left) and anti-SPTLC2 antibodies (right).
Source data are available for this figure.

Together, the co-immunoprecipitation of SPTLC1 with mitochondrial SPTLC2 and their assembly into a complex show that SPT can assemble across organelles. Structural analysis of human SPT reveals how the aminotransferase domains of SPTLC1 and SPTLC2 interact head-to-head to form a dimer with the active site in between the subunits (Li et al, 2021; Wang et al, 2021). Both SPTLC1 and SPTLC2 have a long loop region in between the aminotransferase domain and the N-terminal membrane-bound helix (Li et al, 2021; Wang et al, 2021), which in theory could allow SPTLC2 to reach out to nearby membranes and form a complex with SPTLC1 in cis and in trans. In fact, the SPTLC2 N terminus determines the localization of the protein, as it alone can target both ER and mitochondria (Fig 1F). Regulatory subunits ORMDL1-3 and ssSPTa-b control SPT activity and acyl-chain specificity (Han et al, 2009; Breslow et al, 2010; Harmon et al, 2013; Green et al, 2021). Structural studies reveal details of the interaction of the regulatory subunits with the SPT-complex, and these protein interactions may also play a role in determining SPT localization and activity.

We aimed to recapitulate the SPT complex formation on native gels in A431 SPTLC2-KO cells expressing stably integrated ER-

and OM-specific SPTLC2[FLAG]. However, the expression levels of ER- or OM-SPTLC2[FLAG] were lower than for WT-SPTLC2[FLAG] (Fig S3D) and these artificially targeted, N-terminally modified constructs did not fully support SPT complex formation, as levels and molecular weights of the complexes on a native gel were reduced compared to WT-SPTLC2[FLAG] expressing cells (Fig S3E). Also, despite interacting with SPTLC1, ER-, and OM-SPTLC2[FLAG] did not support sphingolipid synthesis in cells (Fig S3F), suggesting that either the expression levels were too low to contribute to cellular de novo sphingolipid synthesis, or SPTLC2 N terminus is required for full SPT acitivity or for the recruitment of interacting proteins important for the activity.

### SPTLC2 controls palmitate-induced mitochondrial fragmentation

Assembly of SPT at ER–mitochondria contact sites and the presence of other enzymes from the de novo sphingolipid synthesis pathway in mitochondria suggest that SPT-synthesized 3-keto-sphinganine

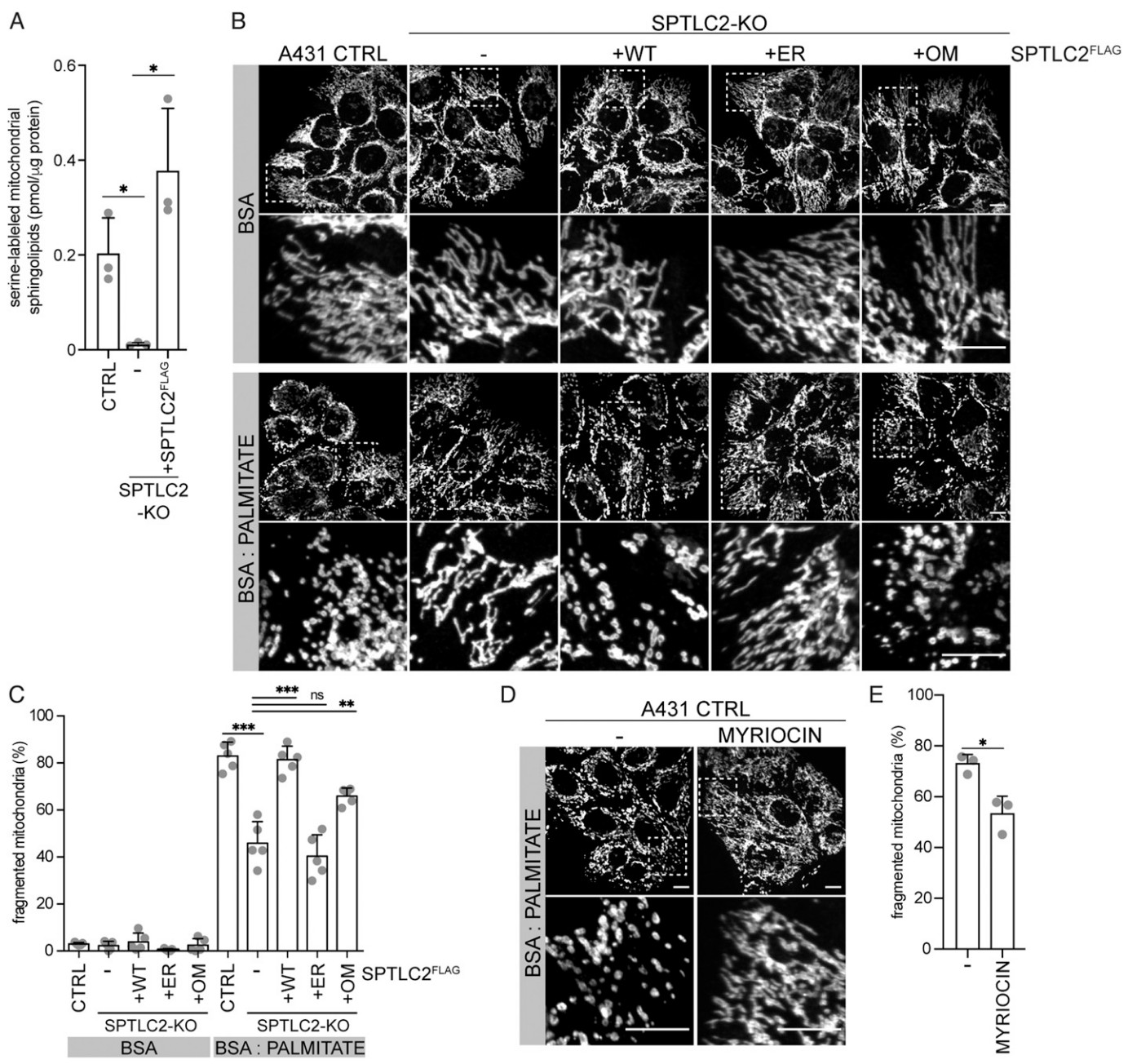

**Figure 4. Lack of mitochondrial SPTLC2 protein attenuates palmitate induced mitochondrial fragmentation.**
**(A)** Quantification of newly synthesized sphingolipids in mitochondria. A431 control, SPTLC2-KO, and SPTLC2-KO cells constitutively expressing WT-SPTLC2[FLAG] were grown in the presence of (2,3,3-D3, [15]N)-L-serine for 24 h. Mitochondria were isolated by gradient-purification and stable isotope-labeled sphingolipids were extracted and analyzed by liquid chromatography–mass spectrometry. Mean ± SD n = 3 independent replicates, unpaired two-sided Welch's $t$ test, *$P < 0.05$. **(B)** Representative confocal images of mitochondrial morphology in A431 cells by TOMM40 immunofluorescence after 8 h control (BSA) or palmitate (BSA:PALMITATE) treatment. A431 control, SPTLC2-KO, and SPTLC2-KO cells constitutively expressing WT-, ER-, or OM-SPTLC2[FLAG]. Scale bar 10 $\mu$m. **(C)** Quantification of cells with a fragmented mitochondrial network. 125–269 cells were counted from 19 to 34 images/genotype/experiment. Mean ± SD n = 5 independent replicates, unpaired two-sided $t$ test, **$P < 0.01$, ***$P < 0.001$, ns, not significant. **(D)** Representative confocal images of mitochondrial morphology in A431 control cells by TOMM40 immunofluorescence after 8 h palmitate treatment in the absence or presence of myriocin. Scale bar 10 $\mu$m. **(E)** Quantification of cells with a fragmented mitochondrial network. 231–367 cells were counted from 26 to 30 images/condition/experiment. Mean ± SD n = 3 independent replicates, unpaired two-sided $t$ test, *$P < 0.05$.
Source data are available for this figure.

could be used for ceramide synthesis in mitochondria. We purified mitochondria (Fig S4A) from A431 cells treated with stable isotope-labeled serine and analyzed the newly synthesized labeled

sphingosine and ceramides by liquid chromatography–mass spectrometry. Mitochondria contained SPT-derived labeled sphingolipids (Fig 4A), with sphingosine and very long-chain $C_{24}$-N-acyl ceramides

and hexosylceramides as the most abundant lipid species identified (Table S3). SPTLC2-KO mitochondria showed an almost complete absence of these lipids which could be rescued by re-expression of SPTLC2[FLAG] (Fig 4A), suggesting that SPT contributes to mitochondrial sphingolipid levels.

In cultured cells, palmitate treatment has been shown to promote mitochondrial fragmentation (Zhang et al, 2010), mimicking the mitochondrial fragmentation induced by ceramide accumulation in liver mitochondria of obese mice where excess fatty acids are provided by a high-fat diet (Hammerschmidt et al, 2019). The fragmentation is related to the mitochondrial accumulation of CerS6-derived $C_{16:0}$ sphingolipids and the binding of these lipids to the mitochondrial fission factor MFF, as CerS6 or MFF deficiency protected the cells from palmitate or diet-induced mitochondrial fragmentation in cultured cells and in vivo (Hammerschmidt et al, 2019). Given the critical role of mitochondrial ceramides in regulating mitochondrial morphology, we assessed the extent of palmitate-induced mitochondrial fragmentation in SPTLC2 deficient A431 cells (Figs S3D and S4B). The treatment of control cells with BSA-conjugated palmitate caused prominent fragmentation of the normally tubular mitochondrial network (Fig 4B and C). SPTLC2-KO cells were resistant to palmitate-induced fragmentation which was rescued upon re-expression of WT-SPTLC2[FLAG]. Although SPT-activity assay in cells suggested that ER- and OM-SPTLC2[FLAG] could not support sphingolipid synthesis (Fig S3F), in the presence of palmitate the expression of OM-SPTLC2[FLAG] lead to a partial restoration of mitochondrial fragmentation, whereas ER-SPTLC2[FLAG] did not rescue the phenotype (Fig 4B and C). Partial rescue by OM-SPTLC2[FLAG] could be explained either by a sufficient mitochondria-localized SPT-activity induced by palmitate or role of SPT as an ER–mitochondria tether supporting alternative routes of sphingolipid transport to mitochondria.

SPT activity is inhibited by myriocin, which blocks the catalytic site of SPT without affecting either the interaction between SPTLC1 and SPTLC2 or integrity of the complex (Wang et al, 2021). Myriocin inhibited palmitate-induced fragmentation (Fig 4D and E), suggesting that palmitate-induced fragmentation depends on SPT catalytic activity rather than a role of the SPT complex tethering at ER–mitochondria contact sites. These results demonstrate that mitochondrial SPTLC2 is a critical regulator of mitochondrial morphology upon fatty-acid challenge, a process dependent on mitochondrial ceramides. Inhibiting mitochondrial ceramide accumulation prevents the development of obesity and insulin resistance in mice (Hammerschmidt et al, 2019). KO of SPTLC1 and SPTLC2 in mice is embryonically lethal (Hojjati et al, 2005); however, pharmacological inhibition of SPT by myriocin in adult rats and mice prevents the development of insulin resistance and weight gain (Holland et al, 2007; Kurek et al, 2014).

This study demonstrates that the SPTLC1 and SPTLC2 subunits can interact in cis on the ER and in trans across membranes to form a functional SPT complex, the latter suggesting that de novo sphingolipid synthesis occurs at ER–mitochondria contact sites (Fig 5). Interaction of two subunits of an enzymatic complex in trans across organelles is highly unusual. Similar mechanisms have been suggested for phospholipid synthesis where an enzyme from the opposing membrane is capable of synthesizing phospholipids in trans at ER–plasma membrane and at mitochondrial inner and

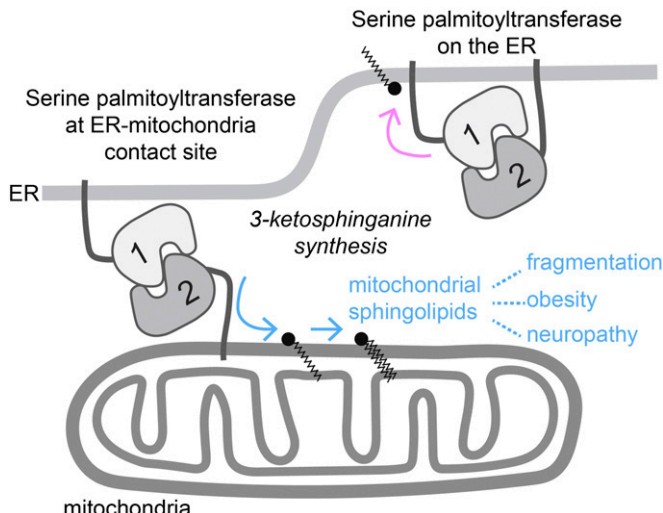

**Figure 5. Model of SPT localization.**
Dual localization of SPTLC2 to the ER and mitochondria directs the SPT complex both to the ER and ER–mitochondria contact sites, respectively. SPTLC2 on the ER interacts in cis with ER-localized SPTLC1, whereas SPTLC2 on the mitochondrial outer membrane interacts in trans with SPTLC1 at membrane contact sites, providing mitochondria with ceramide precursors. SPTLC2 sensitizes cells to palmitate-induced fragmentation, and mitochondrial ceramide accumulation is linked to development of obesity and insulin resistance. Pathogenic SPT variants associated with neuropathy promote deoxy-sphingolipid synthesis.

outer membrane contact sites (Tavassoli et al, 2013; Aaltonen et al, 2016). In trans assembly of SPT at membrane contact site could provide a local source of 3-keto-sphinganine for ceramide synthesis on mitochondria and MAMs, as we and others have observed that the other enzymes required for de novo sphingolipid synthesis are also present in mitochondria, and we show here that de novo synthesized sphingolipid species reach mitochondria. In addition, multiple lipid synthesis and transport steps may co-exist at ER–mitochondria contact sites to ensure mitochondrial sphingolipid supply. Variants of SPTLC1 and SPTLC2 associated with hereditary sensory and autonomic neuropathy are pathogenic because of their ability to generate harmful deoxy-sphingolipids (Gable et al, 2010; Penno et al, 2010; Bode et al, 2016), which have been shown to promote mitochondrial dysfunction in vitro (Alecu et al, 2017; Wilson et al, 2018). Mitochondrial SPTLC2 could promote the localization of endogenous deoxy-sphingolipids to mitochondria, possibly explaining the pathogenesis of the disease.

# Materials and Methods

### Cell culture

Flp-In T-REx 293 (Invitrogen), A431 (ATCC CRL-1555), COS-7 (ATCC), and Phoenix packaging (a kind gift of Garry P Nolan) cell lines were grown in high-glucose DMEM (Wisent 319-027-CL) supplemented with 10% fetal bovine serum in 5% $CO_2$ incubator at 37°C. Cell lines were regularly tested for mycoplasma contamination. For immunofluorescence analysis, cells were cultured on glass coverslips.

For palmitate treatment of A431 cells, 500 $\mu$M palmitate (P0500; Sigma-Aldrich) in ethanol was conjugated to 1% (wt/vol) fatty acid–free BSA (Bioshop ALB006) in DMEM (with all supplements) at 37°C for 20 min. A431 cells grown to 50–70% confluency on coverslips in normal DMEM, were treated with BSA:palmitate or BSA:EtOH for 8 h at 37°C. For inhibition of SPT, 1 $\mu$g/ml Myriocin (M1177; Sigma-Aldrich) in MetOH, or MetOH as control, was added to the cells when indicated. Cells were fixed with pre-warmed 4% paraformaldehyde in PBS for 20 min at 37°C and subjected to immunofluorescence.

## Generation of cell lines

KO of SPTLC1 and SPTLC2 was generated by CRISPR-Cas9–mediated gene editing in Flp-In T-REx 293 and A431 cells. Gene-specific target sequences (SPTLC1: TCTCTAAAGAAGTATGGCGT; SPTLC2: GTTGCTGTG CTCACGTATGT) were cloned into pSpCas9(BB)-2A-Puro (PX459) V2.0 (62988; Addgene) (Ran et al, 2013) and transfected into cells by Lipofectamine 3000 (Thermo Fisher Scientific) according to the manufacturer's instructions. The day after, transfected cells were selected by addition of puromycin (2.5 $\mu$g/ml) for 2 d. Clonal cells were screened for loss of target protein by immunoblotting and frameshift mutations were confirmed by genomic sequencing. Flp-In T-REx 293 KO cells were rescued by integration of tetracycline inducible SPTLC1 or SPTLC2[FLAG] on pDEST-pcDNA5 plasmids as described before (Antonicka et al, 2020). Protein expression was induced with 1 $\mu$g/ml tetracycline for 16–20 h. A431 and U2OS KO cells were rescued by retroviral infection of virus produced in Phoenix cells transfected with pBABE-Puro plasmids as described before (Antonicka et al, 2020).

## Confocal microscopy and stimulated emission depletion (STED) super-resolution microscopy

For confocal microscopy, COS-7 cells were transfected using Lipofectamine 3000 with the indicated pcDNA5 or ER-marker plasmids, fixed the day after with 6% formaldehyde in PBS, solubilized with 0.1% Triton X-100 in PBS for 15 min at RT, and blocked with 5% (wt/vol) BSA in PBS. Primary and secondary antibodies were diluted in 5% (wt/vol) BSA in PBS, and primary antibodies were incubated 3 h at RT or overnight at 4°C, and secondary antibodies were incubated 1 h at RT. Image acquisition (z-stacks, step size 0.2 $\mu$m) was performed with an Olympus IX83 confocal microscope containing a spinning disk system (Andor/Yokogawa CSU-X) using Olympus UPLSAPO 100× or PLAPON 60× oil objectives. Images were processed in Fiji (Schindelin et al, 2012).

For STED microscopy, cells were transfected with ER-mEmerald plasmid using Lipofectamine 3000 and plated on high-precision cover glasses (ES0117520; Azer Scientific). Cells were fixed with 6% paraformaldehyde in PBS, quenched with 50 mM NH$_4$Cl/PBS, pH 7.4, permeabilized with 0.05% Triton X-100 in PBS for 10 min with three PBS washes in between steps. Cover glasses were blocked with 10% FBS (vol/vol in PBS) and incubated with primary and secondary antibodies with five washes with 5% FBS in between steps. Cells were re-fixed with 6% paraformaldehyde for 30 min at 37°C and quenched with 50 mM NH$_4$Cl before mounting. STED super-resolution microscopy was performed using an Abberior Expert Line STED microscope with two pulsed STED lasers (595 and 775 nm) based on

an Olympus IX83 inverted microscope with an Olympus Plan-Apo 100×/1.40 NA oil objective and a pixel size of 25 nm.

## Plasmids

ER-mCherry (mCh-Sec61 beta) plasmid was a gift from Gia Voeltz (49155; Addgene) (Zurek et al, 2011). ER-mEmerald (mEmerald-Sec61b-C1) plasmid was a gift from Jennifer Lippincott-Schwartz (90992; Addgene) (Nixon-Abell et al, 2016). SPTLC1 (untagged or single C-terminal FLAG-tag incorporated in primer) and SPTLC2 (full length or truncated) coding sequences were amplified from cDNA and cloned into pDONR221 (Invitrogen) using Gateway cloning technology. C-terminally GFP-tagged SPTLC2 variants were cloned into pcDNA-pDEST47 Gateway destination vector (Invitrogen). ER- and OM-SPTLC2 were cloned by first adding XbaI and XhoI sites in front of SPTLC2$_{103-562}$ by mutagenesis PCR, then SPTLC1$_{1-49}$ or mAKAP1$_{1-31+LINKER}$ (MAIQLRSLFPLALPGLLALLGWWWFFSRKKDLERKKGSKPGSK) coding sequences were introduced into the site. pDONR221 vectors were flipped into destination vectors by Gateway technology. SPTLC1 was flipped into pDEST-pcDNA5 empty vector, which was generated by deleting BirA*-FLAG sequence from pDEST-pcDNA5-BirA*-FLAG C-term (Couzens et al, 2013) by XbaI digestion. SPTLC2 was cloned into pDEST-pcDNA5-3xFLAG vector, which was generated by cloning the 3xFLAG peptide sequence (DYKDHDGDYKDH-DIDYKDDDDK) into pDEST-pcDNA5 empty vector with ApaI. Gateway compatible pBABE-3xFLAG vector was generated by cloning the Gateway cassette and 3xFLAG sequence from plasmid pCSF107mT-Gateway-3'-Flag (a gift from Todd Stukenberg, 67619; Addgene) with blunted BglII and XhoI sites into SnaBI site of pBABE-Puro (1764; Addgene) (Morgenstern & Land, 1990).

## qRT-PCR

Total RNA was isolated from cells using miRNeasy kit (QIAGEN). qRT–PCR analysis for *SPTLC1* (primers: ttcctcctgtcccaaaagac, cac-cacagttttgtggcttg) and *SPTLC2* (primers: acggaacgggtacgtgag, tttg tgtaacatgatggatctgg) mRNAs was performed at the Institute for Research In Immunology and Cancer (IRIC).

## Subcellular fractionation

For preparation of mitochondria enriched heavy membrane fractions, cells were resuspended in isolation buffer IB (220 mM mannitol, 70 mM sucrose, 20 mM HEPES-KOH, pH7.6, 1 mM EDTA, 1× cOmplete protease inhibitor [PI, Roche 11873580001; for protease protection assay, EDTA and PI were omitted]) and homogenized by 10–15 strokes in a rotating Teflon-glass homogenizer at 1,000 rpm. Homogenates were centrifuged twice at 800$g$ for 5 min to remove nuclei and cell debris. Heavy membranes were pelleted at 8,000$g$ for 5 min, washed with buffer IB, resuspended in buffer IB, and protein concentration was determined by Bradford assay.

For subcellular fractionation of A431 cells or SPTLC2[FLAG] expressing Flp-In T-REx 293 cells, cells (4 × 15 cm dishes for 293, 6 × 15 cm dishes for A431) were washed with PBS and resuspended in buffer M (220 mM mannitol, 70 mM sucrose, 5 mM Hepes/KOH, pH 7.4, 1 mM EDTA, and 1× PI). The cell suspension was homogenized by 15 strokes with a rotating Teflon-glass homogenizer at 1,000 rpm

followed by differential centrifugation. The homogenate was centrifuged twice at 600$g$ for 5 min to remove the debris and nucleus, and the resulting supernatant was then centrifuged at 8,000$g$ for 10 min to obtain mitochondria enriched heavy membrane fraction. The resulting supernatant was first centrifuged at 8,000$g$ to remove remaining mitochondria, and then centrifuged at 100,000$g$ for 30 min to separate cytosolic (supernatant) and light membrane/microsome fractions (pellet). For further mitochondrial purification, mitochondria enriched heavy membranes were suspended in buffer M and purified by density gradient centrifugation over a 40% (2 ml)–19% (4 ml)–12% (4 ml) Percoll gradient (42,000$g$, 30 min, SW40Ti rotor). Fractions 1, 2, and 3 were collected from top to bottom, diluted with 1:5 with isolation buffer, and washed three times with isolation buffer (20,000$g$, 10 min). All pellets were resuspended into buffer M, and protein concentrations were determined by Bradford.

Fractionation of whole liver was performed as described before (Wieckowski et al, 2009) with small modifications. The C57/Bl6N male mouse was purchased from Jackson Laboratories, and liver harvesting and animal handling were approved and performed in accordance with the Montreal Neurological Institute Animal Care Committee regulations. 10-mo-old male mouse was starved overnight, euthanized, and the whole liver was resected and washed in ice-cold buffer MIB (225 mM mannitol, 75 mM sucrose, 20 mM Tris, pH 7.4, 0.5 mM EGTA, and 0.5% [wt/vol] BSA). The liver was homogenized in 10 ml of buffer MIB by 10 strokes in a rotating Teflon-glass homogenizer at 1,000 rpm. The homogenate was centrifuged at 740$g$ for 5 min, and the pellet was re-homogenized and centrifuged at 740$g$ for 5 min. The homogenates were combined and centrifuged twice at 740$g$ for 5 min to remove debris and nucleus, and post-nuclear homogenate sample was collected from the resulting supernatant. To pellet mitochondria-enriched heavy membranes, the supernatant was centrifuged at 9,000$g$ for 10 min. The resulting supernatant was centrifuged five times at 10,000$g$ for 10 min to remove remaining heavy membranes, and then centrifuged at 100,000$g$ for 30 min to separate cytosol (supernatant) and light membranes/microsomes (pellet). Mitochondria-enriched heavy membrane pellet was washed twice with buffer MRB (250 mM mannitol, 5 mM Hepes [pH 7.4], 0.5 mM EGTA, and 1× PI) by resuspending the pellet into 10 ml MRB and centrifuging 10,000$g$ for 10 min, and resuspended into 2.5 ml MRB from which the heavy membrane sample was collected. To separate mitochondria and MAMs, heavy membranes were purified by density gradient centrifugation over Percoll medium (225 mM mannitol, 25 mM Hepes, 1 mM EGTA, 30% Percoll [vol/vol]; 8 ml Percoll medium, 2 ml sample, and 3.5 ml MRB) at 95,000$g$ for 30 min (SW40Ti rotor). Mitochondria were collected from the bottom part of the tube and diluted 1:10 with buffer MRB, pelleted at 6,400$g$ for 10 min, and washed twice with buffer MRB. MAMs were collected from the middle of the tube, diluted 1:10 with buffer MRB, centrifuged twice at 6,400$g$ to remove residual mitochondria and pelleted at 100,000$g$ for 30 min. All pellets were resuspended into buffer MRB, and protein concentrations were determined by Bradford.

**Protease protection assay**

Mitochondria enriched heavy membrane pellets (100 μg) were resuspended in buffer IB (without EDTA and PI), or 20 mM HEPES-KOH, pH 7.4, or 1% (vol/vol) Triton X-100 in 20 mM Hepes, pH 7.4. Proteinase K (50 μg/m, P2308; Sigma-Aldrich) was added and samples were incubated for 15 min on ice. Protease was inactivated by 2 mM PMSF for 5 min on ice. Samples were subjected to TCA precipitation and pellets were dissolved into Laemmli buffer.

**Alkaline carbonate extraction**

Mitochondria enriched heavy membrane pellet (100 μg) was resuspended in alkaline carbonate (100 mM $Na_2CO_3$, pH 11.5) and incubated 30 min on ice. Half was set aside for total input sample and half was ultracentrifuged at 100,000$g$ for 30 min. Supernatant and total samples were subjected to TCA precipitation, membrane pellet was rinsed with alkaline carbonate, and pellets were dissolved into Laemmli buffer.

**Immunoprecipitation and mass spectrometry analysis**

Heavy membrane fractions were solubilized in buffer (50 mM TRIS, pH 7.4, 150 mM NaCl, 1% 4 g digitonin [EMD Millipore #300410]/g protein [1% wt/vol], and 1× PI) for 20 min and the aggregates were pelleted at 20,000$g$ for 15 min. The supernatant was incubated with anti-FLAG magnetic M2 beads (M8823; Sigma-Aldrich) for 3 h. For ms analysis, the samples were eluted using 100 mM glycine, pH 2.5, TCA precipitated, and trypsin digest and mass spectrometry analysis was performed at the Institute de Recherches Cliniques de Montreal. For immunoblot analysis samples were eluted with Laemmli buffer. Information for subcellular and sub-mitochondrial localization in Table S2 were retrieved from UniProt Consortium (2021) and Mito-Carta3.0 (Rath et al, 2021), respectively.

**Denaturing and native PAGE**

For analysis of protein levels in whole cell lysates, cells were lysed with RIPA lysis buffer (1% Triton X-100, 0.1% [wt/vol] SDS, 0.5% [wt/vol] sodium deoxycholate, 1 mM EDTA, 50 mM TRIS pH 7.4, 150 mM NaCl, and 1× PI) for 20 min and centrifuged for 15 min at 20,000$g$. Equal amounts of protein were analyzed on 10% Tris-Tricine SDS–PAGE system (Schagger & von Jagow, 1987) with Precision plus protein standard (1610363; Bio-Rad) as a molecular weight marker, and blotted onto a nitrocellulose membrane. Protein intensities were quantified on Fiji. For analysis of SPT complex on blue-native PAGE, mitochondria enriched heavy membrane fractions were solubilized with 4 g digitonin/g protein (1% wt/vol) for 20 min, centrifuged for 20 min at 20,000$g$ and 15 μg of supernatants were separated on a 4–13% gradient gel as previously described (Leary, 2012) with Amersham HMW native marker kit (17044501; GE Healthcare) as a molecular weight marker, and blotted onto a polyvinylidene difluoride (PVDF) membrane.

**Sphingolipid labeling assay and lipid analysis**

Cells were grown to 70% confluency in regular DMEM. For labeling, cells were first serine-starved for 2 h in L-serine-free DMEM (Wisent 319-130) containing 4.5 g/l glucose and 10% dialyzed FBS, and then grown for 24 h in the presence of 1 mM (2,3,3-D3, $^{15}$N)-L-serine (DNLM-6863; Cambridge Isotope Laboratories). Myriocin (1 μg/ml) in MetOH was added when indicated. Cells were harvested on ice,

counted and cell pellet was collected for whole cell lipid analysis, or mitochondria were isolated by gradient-purification.

Sphingolipids were extracted on ice using a modified Bligh and Dyer method, as previously described (Xu et al, 2013). Briefly, samples were transferred to glass Kimble vials and 3.7 ml of methanol (# BP1105-4; Thermo Fisher Scientific) acidified with 2% acetic acid (Cat. no. A38-212; Thermo Fisher Scientific) was added to each sample. Deuterated lipid standards—Cer(d18:1/16:0-D31) #868516, GlcCer(d18:1/8:0) #860540, GalCer(d18:1/8:0), #860538, and SM(d18:1/18:1-D9)# 791649 (Avanti Polar Lipids)—were added at the time of extraction, followed by chloroform (#C298-500; Thermo Fisher Scientific) and 0.1 M sodium acetate (JT Baker, #9831-03) to a final ratio with acidified methanol of 2:1.9:1.6, respectively. Samples were vortexed after each step and then centrifuged for 5 min at 4°C and 600$g$. The organic phase was collected into a new tube, and the aqueous phase was back-extracted an additional three times using chloroform, with each subsequent organic phase collected being pooled with the previous one. The samples were dried under a constant stream of nitrogen, and lipids were re-solubilized in 300 $\mu$L of 100% ethanol (Commercial Alcohols, P016EAAN), flushed with nitrogen, and stored at –80°C.

The heavy isotope-labeled sphingolipids (m/z + 3, one of the deuterium labels in serine is lost upon condensation with palmitoyl-CoA) were analyzed by high-performance liquid chromatography–electrospray ionization tandem mass spectrometry (LC-ESI-MS/MS) on an Agilent 1290 Infinity II liquid chromatography system coupled to a QTRAP 5500 triple quadrupole-linear ion trap mass spectrometer with Turbo V ion source (AB SCIEX). Reverse-phase chromatography was performed using a binary solvent gradient with solvent A (water with 0.1% formic acid [Fluka, #56302] and 10 mm ammonium acetate [OmniPur, #2145]) and solvent B (acetonitrile [JT Baker, #9829-03] and isopropanol [#A461-4; Thermo Fisher Scientific] at a ratio of 5:2 vol/vol with 10 mm ammonium acetate and 0.1% formic acid) pumped over a 100-mm × 250-$\mu$m (inner diameter) capillary column packed with Reprosil-Pur 120 C8 at a flow rate of 10 $\mu$l/min. The duration of the method was 50 min, with the gradient starting at 30% B and reaching 100% B over the first 5 min. This solvent compostion was maintained until 35 min, and then ramped down to 30% B by 36 min and maintained until the end of the run. The mass spectrometer was run in positive ion mode using selected reaction monitoring, monitoring protonated molecular ions of the full molecular mass of m/z + 3 sphingolipid species in Q1, and m/z 267.3 or 269.3 (sphingosine or sphinganine backbone +3, respectively) in Q3.

Analyst software version 1.6.2 (AB SCIEX) was used for acquiring data and MultiQuant 3.0.2 software version 3.0.8664.0 (SCIEX) was used for quantification. Raw peak areas were normalized to cell number or protein concentration, as well as to the appropriate internal standards added at the time of extraction to account for extraction efficiency and instrument response.

### Antibodies

The following antibodies were used for immunofluorescence analysis: mouse anti-FLAG (F1804; Sigma-Aldrich), rabbit anti-FLAG (20543-1-AP; Proteintech), anti-MRPL12 (H00006182-M01; Abnova), anti-GFP Alexa Fluor 488 (A-21311; Invitrogen), anti-SPTLC1 (Hornemann et al, 2006), anti-DEGS1 (HPA076422; Sigma-Aldrich),

anti-TOMM20 (HPA011562; Sigma-Aldrich), anti-TOMM40 (18409-1-AP; Proteintech), anti-CANX (699401; BioLegend), anti-PRDX3 (Jeyaraju et al, 2006), anti-CYCS (556432; BD Biosciences), and the secondary antibodies anti-rabbit Alexa Fluor 488 (A-21206; Invitrogen), anti-mouse Alexa Fluor 647 (A-31571; Invitrogen), anti-rat Alexa Fluor 594 (A-11007; Invitrogen), Abberior STAR 580 goat anti-rabbit IgG (ST580-1002; STED) and Abberior STAR 635P goat anti-mouse IgG (ST635P-1001; STED). The following antibodies were used for protein detection on immunoblot: anti-SPTLC1 and anti-SPTLC2 were a gift from Hornemann et al (2006), anti-CYCS (556433; BD Biosciences) anti-IP3R1 (8568; Cell Signaling Technology), anti-SIGMAR1 (15168-1-AP; Proteintech), anti-PDI (610946; BD Biosciences), anti-UBB (3933; Cell Signaling), anti-KDSR (NBP2-14962; Novus), anti-DEGS1 (HPA076422; Sigma-Aldrich), anti-MFN2 (11925; Cell Signaling), anti-CHCHD4 (21090-1-AP; Proteintech), anti-LRPPRC (166178; Santa Cruz Biotechnology), anti-$\beta$-ACTIN (A00702; GenScript), anti-ATP5A (ab14748; Abcam), anti-TOMM20 (11415; Santa Cruz Biotechnology), anti-MTCH2 (16888-1-AP; Proteintech), anti-GK (13360-1-AP; Proteintech), anti-CLPB (HPA039005; Sigma-Aldrich), anti-SYNJ2BP (15666-1-AP; Proteintech), anti-SDHA (ab14715; Abcam), anti-GRSF1 was a gift from J Wilusz (Qian & Wilusz, 1994), anti-VAPB (14477-1-AP; Proteintech), anti-MRPL44 (16394-1-AP; Proteintech), anti-SPTLC2 (used in Fig 1E, Novus NBP1-76573), and the secondary antibodies anti-mouse-HRP (115-035-146; Jackson ImmunoResearch), anti-rabbit-HRP (111-035-003; Jackson ImmunoResearch).

### Statistical analysis

Data are expressed as mean ± SD. Statistical evaluation was performed using unpaired $t$ test. Graphs and statistical analysis were generated with GraphPad Prism 8.

# Data Availability

All data are available in the supplementary Tables (Tables S1–S3). This study includes no data deposited in external repositories.

# Supplementary Information

# Acknowledgements

We thank Kathleen Daigneault for technical assistance, and the members of the Shoubridge lab for helpful discussions. We thank Aurèle Besse-Patin and Heidi McBride for the mouse liver, and Michiel Krols for helpful discussions. We thank Thorsten Hornemann for sharing the SPTLC1 and SPTLC2 antibodies. This research was funded by grants from Canadian Institues of Health Research to EA Shoubridge (133530) and SAL Bennett (163902). MJ Aaltonen was supported by a postdoctoral fellowship from the Healthy Brains Healthy Lives initiative. T König was supported by a postdoctoral fellowship from the Alexander von Humboldt Foundation. I Alecu was supported by a Parkinson's Research Consortium Crabtree Family post-doctoral fellowship.

## Author Contributions

MJ Aaltonen: conceptualization, formal analysis, validation, investigation, methodology, and writing—original draft, review, and editing.
I Alecu: formal analysis, investigation, methodology, and writing—review and editing.
T König: resources, investigation, and methodology.
SAL Bennett: formal analysis, investigation, methodology, and writing—review and editing.
EA Shoubridge: conceptualization, formal analysis, supervision, funding acquisition, methodology, and writing—original draft, review, and editing.

## Conflict of Interest Statement

The authors declare that they have no conflict of interest.

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
