## [Reviewer comments · Life Science Alliance]

Serine Palmitoyltransferase Assembles at ER-Mitochondria Contact Sites

Mari Aaltonen, Irina Alecu, Tim König, Steffany Bennett, and Eric Shoubridge

DOI: <https://doi.org/10.26508/lsa.202101278>

Corresponding author(s): Eric Shoubridge, McGill University

Review Timeline:	Submission Date:	2021-10-25
	Editorial Decision:	2021-10-26
	Revision Received:	2021-10-29
	Editorial Decision:	2021-11-01
	Revision Received:	2021-11-01
	Accepted:	2021-11-03

Transaction Report:

Referee #1 Review

Report for Author:

Please note that, as asked by the editor, in my answer I will take in account also the points raised by reviewer 2.

I do appreciate the improvements and the new set of experiments presented in this revised version, including those in KO cells, that reinforce the function of SPTLC1 and SPTLC2 in regulating mitochondrial sphingolipid synthesis. However, the authors propose that SPTLC1 and SPTLC2 control palmitate-induced mitochondrial fragmentation (via direct sphingolipid synthesis on mitochondria) by interacting "in trans" at ER-mitochondrial contact sites (with SPTLC1 being mainly localized in the ER and SPTLC2 in both ER and mitochondria).

Although this hypothesis is very novel and intriguing, the overall data support both the models where SPTLC1 and SPTLC2 could interact "in cis" and "in trans". Unfortunately, to my opinion the provided experiments are not sufficient to specifically validate the model where SPTLC1 and SPTLC2 interact and function "in trans" at ER-mitochondria junctions nor the functional relevance and existence of the "in trans" interaction in physiological conditions. In other words, a clear evidence that the observed effects upon their knockdown depend on their interaction "in trans" (rather than "in cis") is still missing.

Specific comments:

- The super-resolution of SPTLC2-FLAG show a uniform diffuse distribution across the ER and consequently in the ER near mitochondria (MAM). However, there is no enrichment in ER close to mitochondria, as it would be expected if SPTLC2-FLAG was behaving similarly to the endogenous SPTLC2 (shown in Figure 1D to be enriched in the mitochondrial fraction). Thus, the super-resolution of SPTLC2-FLAG is not particularly supporting the "in trans" model.
- The occurrence of SPTLC1- SPTLC2 "in trans" interaction has been shown using artificially targeted constructs (ER-SPTLC2, exclusively targeted to ER membranes, and OM-SPTLC2, exclusively targeted to mitochondrial membranes). However, these experiments do not prove the existence of a physiological interaction "in trans" between SPTLC1- SPTLC2 at ER-mitochondrial contacts. Also, the native PAGE experiment presented here to support the authors "in trans" model could be interpreted differently: the absence of the complex in the pure mitochondria fraction (and presence of only monomeric SPTLC2) could also be explained by the loss of the main pool of SPTLC1- SPTLC2 interacting "in cis" in the ER, that is not detectable in the purified mitochondria.
- It is possible that the observed phenotypes in KO cells are due to the loss of SPTLC1- SPTLC2 interacting "in cis" in the MAM at ER-mitochondria contacts. SPTLC1- SPTLC2 interacting "in cis" could form a macromolecular complex with several mitochondria proteins that might contribute to their localization at MAM.

- Quantitative aspects are still missing (see Reviewer 2 comments): no comparison of the specific activities of SPTLC1- SPTLC2 assembled in cis versus SPTLC1- SPTLC2 assembled in trans, has been shown.

Minor:

- It would have been good to show co-localization of SPTLC1- SPTLC2 at ER-mitochondria contacts, even in co-overexpression (given the lack of endogenous antibody working for IF for SPTLC2).
- Page 6 line 13: "line" instead of "ine".

Referee #3 Review

Report for Author:

Comments to 2021-52837V2

I found the manuscript has been greatly improved by the revision and I was almost satisfied with the quality of work presented. However, the current manuscript has a few minor concerns, which are listed below. I consider these concerns should be removed before acceptance.

1) p3, Fig. 1A, and others: Although sphingaine may be a convenient term to represent an isoform of sphingoid bases, dihydrosphingosine is a more systematic nomenclature to represent the sphingoid base isoform, compared with sphinganine (e.g., Hanada, BBA, 2003, 1632, 16-30). Thus, the term of sphingaine should better be changed to dihydrosphingosine throughout the manuscript. KDSR is after 3-**keto-dihydro**sphingosine reductase, not 3-keto-sphingaine reductase (p4). Of note, dihydroceramide terminologically means N-acylated dihydrosphingosine, not N-acylated sphinganine.

2) p13, 1% (4 g/g) digitonin: what does (4 g/g) mean? Do the authors mean 1% (weight/weight)? The manufacturer information should be added because digitonin is a crucial reagent in this study.

3) As per the response of the authors to my major comment #1: Although I agreed with the answer, the brief explanation should be made in the text for audiences. For example, the following sentences by the authors' answer may be added: "As both SPTLC1 and SPTLC2 are required for SPT complex formation and thus SPT activity, the problem with performing activity assays on gradient-purified mitochondria and MAMs is that there will not be any enrichment of activity due to only one half of the enzyme complex being present in each fraction."

Referee #1 Review

Report for Author:

The accumulation of sphingolipids in the cell is linked to obesity and neurological disease. However, the subcellular localization of sphingolipid synthesizing enzymes is still unclear, limiting the understanding of where and how these lipids accumulate inside the cell and why they are toxic.

In this manuscript, Aaltonen and Shoubridge show that SPTLC2, a subunit of the serine palmitoyltransferase (SPT) complex catalyzing the first step of de novo sphingolipid synthesis, when overexpressed (-GFP and -FLAG tagged) localizes dually to the endoplasmic reticulum (ER) and to the outer mitochondrial membrane. They also show that the mitochondrial SPTLC2 interacts and forms a complex with the ER-localized SPT subunit SPTLC1. Lastly, they show that SPTLC2 depletion protects cells from palmitate-induced mitochondrial fragmentation, a process that can be regulated by ceramides. They propose that the SPTLC2-SPTLC1 enzymatic complex assemble in trans at ER-mitochondria membrane contact site to regulate synthesis of sphingolipids directly on mitochondria.

The novel localization of SPTLC2 on mitochondria, that is the main novel finding of this manuscript, is very interesting as it suggests that indeed a sphingolipid biosynthetic activity could exist at ER-mitochondria contact sites, acting directly on mitochondria. However, the microscopy and biochemical experiments presented here are overall not sufficient, and sometimes mis-interpreted, to define the precise localisation of the SPTLC2- SPTLC1 complex and their physiological relevance at ER-mitochondria contact sites.

Major points:

1. Localization of SPTLC2:

The co-localization of SPTLC2 and SPTLC1 at ER-mitochondria contact sites in co-expression and at endogenous levels has

not been analysed. Also, there are some incongruences between experiments shown in Fig 1, suggesting that the additional localization of SPTLC2 to the ER could be just an effect of its individual overexpression (not in accord with its endogenous distribution in the MAM-mitochondria fractions- see below).

More in detail:

- SPTLC2 and SPTLC1 localization is here mostly addressed by using tagged constructs (GFP- and FLAG). The authors say that while SPTLC1 is localized in the ER, SPTLC2 is "present in both ER membranes and mitochondrial outer membrane". However, the data presented do not unanimously support this dual localization. Overexpressed SPTLC2-GFP seems localized mainly to mitochondria rather than in the ER. It is possible that the additional localization of SPTLC2-GFP to the ER reflect a mistargeting due to the tag or to the high expression levels.

The only attempt to address the localization of endogenous SPTLC2 and SPTLC1 is a gradient-purification of MAMs and mitochondria (Fig1D and S2A). Unfortunately, the purified fractions are not pure, as shown by the presence of mitochondrial proteins (i.e. cytochrome c) in the MAM fraction and of ER (PDI) and MAM (IP3R1, VAP) proteins in the mitochondrial fractions. The predominance of the endogenous SPTLC2 in the mitochondrial fraction suggest that this is its main physiological localization while the very weak band of SPTLC2 in the MAM fraction is most likely a contamination, as compared to the cytochrome c, also present at similar levels as SPTLC2 in the MAM. Likewise, SPTLC1 is enriched in the MAM fraction, but also weakly present in the mitochondria fraction, at similar levels as other ER and MAM contaminants.

The other assays used to study SPTLC2 and SPTLC1 association to membranes do not rule out this issue as well, given that they are performed on mitochondrial enriched heavy membrane fractions that contains also MAM.

The localization of endogenous SPTLC2 and SPTLC1 at ER, MAM and/or mitochondria should be convincingly addressed using biochemical approaches that ensure isolation of high-purity fractions and by immunofluorescence/microscopy to prove that their interaction occurs indeed at ER-mitochondria contact sites.

- The localization of SPTLC2 and SPTLC1 has been addressed only when expressed independently and thus not at similar levels. If they are part of a complex they should co-localize at ER-mitochondria contact sites when co-expressed. However, there is no microscopy data showing this.

Also, do the endogenous SPTLC2 and SPTLC1 interact? The interaction between endogenous SPTLC2 and SPTLC1 should be characterized, for example by co-IP and by PLA coupled with mitochondria and ER staining. This will allow to localise SPTLC2-SPTLC1 interaction at the ER-mitochondria interface and also to discriminate between the interaction between SPTLC2-SPTLC1 "in cis" (that is claimed but not clearly proved) versus their interaction "in trans" (that is more functionally relevant) at ER-mitochondria contacts (see below).

2. Characterization of KO cells and protein interactions:

The main finding shown in this section is the interaction between SPTLC2 and SPTLC1, and their reciprocal regulation of their expression levels. However, there is no real functional characterization of the SPTLC2 and SPTLC1 KO cells. Also the interactions with the newly identified ER and mitochondrial binding partners have not been characterized, and localization and functional analyses of these interactions are missing.

Thus, the results shown in this section do not reflect the functional deepness announced by the title. This section is quite redundant with (and thus could be fused to) the following one.

3. The title of this section is: "mitochondrial SPTLC2 interacts in trans with SPTLC1".

Nevertheless, in the text the authors say that SPTLC2- SPTLC1 can interact both in cis and in trans membranes. They arrive to this conclusion by generating FLAG-tagged SPTLC2 constructs that are "artificially" targeted to either ER or mitochondria by replacing the N-terminus of SPTLC2 with the N-terminus of SPTLC1 or AKAP1 (mitochondrial protein) and analysing their interaction with SPTLC1 in comparison with the interaction with TOMM20 or TOMM40 (Fig 3A). SPTLC1 is found in all IPs conditions. TOM20 and TOM40 are not found in the ER-SPTLC2, exclusively targeted to the ER, and are strongly decreased in the OM-SPTLC2, exclusively localized to mitochondria (Fig 3B). They use these results to propose that SPTLC2- SPTLC1 can interact in cis (ER) and in trans (ER-mitochondria membranes). However, to me it seems that the experiments showed here just confirm that these two proteins indeed interact but do not discriminate the precise sites of their interaction or prove that it occurs at contact sites.

Also, why the interaction with the mitochondrial proteins TOM20 and TOMM40 proteins is decreased in the OM-SPTLC2 immunoprecipitates?

The authors say that : "in trans assembly of SPT at membrane contact sites would provide an elegant way to completely avoid lipid transfer from ER by specialised proteins, and allow sphingolipid synthesis directly on mitochondria". However, they do not provide any experimental evidence supporting what is mentioned here.

Despite the importance of having two subunits of an enzymatic complex interacting at the ER-mitochondria interface, where is the evidence that this would exclude the involvement of any other lipid transport pathways at these sites? To my opinion, multiple lipid transfer and biosynthetic pathways could co-exist at ER-mitochondria contact sites.

Also, it is not so clear if the authors refer here to the transfer of specific lipids (sphingolipid precursors) or of lipids in general?

4. The authors show that SPTLC2 regulates palmitate-induced mitochondria fragmentation as SPTLC2 KO cells were resistant

to palmitate-induced fragmentation of mitochondria and this phenotype was rescued by re-expression of WT- SPTLC2 (Fig 4). However, it is not addressed whether the rescue by WT-SPTLC2 re-expression involves its interaction with SPTLC1 at ER-mitochondrial contact sites. At least the authors should show that re-expression of WT- SPTLC2 in a double SPTLC2- SPTLC1 KO is not sufficient to rescue the fragmentation phenotype. Also, the mitochondrial fragmentation might be due to indirect effects on the extent of ER-mitochondria contact sites. Thus, it should be carefully addressed whether these contacts are perturbed (for example decreased) in SPTLC2 KO cells.

The authors propose that SPTLC2 could promote the localization of endogenous deoxy-sphingolipids to mitochondria. However no experiment has been done to address this possibility.

A construct lacking the lipid biosynthesis activity could be used in rescue experiments to support this hypothesis. Also, lipidomics analysis of the sphingolipid content of mitochondria would be required to confirm it.

Minor points:

The partial rescue by OM-SPTLC2 could be explained by its reduced interaction with SPTLC1 (as shown in figure 3B).

Referee #2 Review

Report for Author:

This manuscript presents a very interesting model whereby SPTLC2 is localized to regions of mitochondria forming ER-mitochondrial junctions and SPTLC1 is localized to the ER. They propose that the active enzyme, which normally requires both subunits, forms in trans between the two compartments. This specifically localized SPT is proposed to feed sphingoid base precursors into ceramide synthesis that is targeted somehow to mitochondria and could then lead to apoptosis. This is a very interesting hypothesis and is supported by chimeric enzyme targeting to the ER and mitochondria, but quantitative aspects and the demonstration that the other elements of what would be needed for this pathway are missing. In my opinion, the following points still need to be addressed.

1. The whole model depends upon spatial and quantitative aspects of SPT activity, however, the quantitative aspects are missing. For instance, we do not know if the different forms of SPT (ER-ER, ER-Mito) have similar specific activities. We do not know if the enzyme assembled in trans has a similar specific activity to that assembled in cis in the ER. If one is much more active than the other this could influence the model. It should be possible to measure the activity in vitro on isolated membranes and measure the amount of enzyme by Western blot from the different conditions to have an idea about the specific activity.
2. As the authors mention, there are other enzymes of the pathway that are required to produce the ceramide which they propose reaches the mitochondria and causes apoptosis. One of these is KDSR. It is important to know the localization of KDSR in their cells. Does it change with the localization of the different SPT constructs? There is no available evidence on whether KDSR can be rate limiting in sphingolipid biosynthesis to my knowledge so to assume that its localization is not important is not a valid assumption. Furthermore, if the ketosphinganine has to be transported from junctions to the ER proper for conversion, where is the advantage of generating it at junctions for ceramide synthesis going into mitochondria.
3. The other critical components of the pathway are the CERS and DES1. Where are they localized? The same logic applies. In contrast to KDSR, ceramide synthases do undergo regulation, by phosphorylation and perhaps other mechanisms. And, not all of them can make the correct ceramides to trigger apoptosis. Therefore, I believe it is important also to localize the relevant ceramide synthase and DES1, again because if diffusion back to the ER is necessary to reach the relevant enzymes then the advantage of localization to junctions is lost.

If parts of the enzymatic machinery are not localized at or concentrated very near the junctions, their model could still be correct, but only if all of the subsequent enzymes are not rate limiting. They have not shown this and there is no evidence of it from the literature. One way to address this would be to vary the amounts of these enzymes in cells, partial knockdowns, overexpression, to see if they change the flux through the pathway. The pathway should be relatively insensitive to these changes if the enzymes are not rate limiting. Some evidence does exist on overexpression of ceramide synthases and this does lead to moderate increases in particular ceramide species, so it may be that these enzymes are rate limiting as well. Therefore, it is very important to localize them carefully.

If diffusion of substrates back to the ER proper is required in their analysis an alternative explanation for their results could be proposed. It could be that the SPT intermembrane complex functions as a particular type of membrane junction and its synthetic activity is not important. There are various ways to address this hypothesis, by co-expressing ER and OM forms with one of them catalytically inactive. These could be tricky and lead to different enzyme specific activities as mentioned above.

The model of SPT serving a "non-enzymatic" function at junctions is not far-fetched because SPT has been shown by the Wattenberg lab to be directly regulated by ceramide (<https://doi.org/10.1074/jbc.RA118.007291>), so if the enzyme is found at junctions and binds ceramides it could act to transfer them between ER and mitochondria.

In summary, these are very interesting results and model, but components are still missing to make it compelling.

Referee #3 Review

Report for Author:

Comments to EMBOR-2021-52837V1

The manuscript by Aaltonen and Shoubridge (hereafter the authors) proposes a new model of the SPT complex, in which the SPTLC1 subunit is bound to the ER (as widely believed) and the SPTLC2 subunit is largely bound to the outer membrane (OM) of mitochondria, which is previously unrecognized, at ER-mitochondria contact sites. This model may invoke a conceptual breakthrough in the research of not only SPT but also more broad membrane-bound enzymes. The manuscript provides sound evidence indicating that at least a part of SPTLC2, even if not all, is associating with the OM of mitochondria. However, the current manuscript lacks the data of the subcellular distribution of "SPT activity", which is essential for the main conclusion. Thus, I consider that the manuscript should be properly amended before acceptance. My major and minor specific comments are listed below.

Major comments:

1. The manuscript shows various data for subcellular localization of SPT subunits. However, the current manuscript lacks the data of the subcellular distribution of SPT activity. Thus, it remains unclear whether mitochondria-bound SPTLC2 is relevant to active SPT. The authors should determine the SPT activity (per mg protein) in subcellular fractions (e.g., obtained in Fig. 1D) and assess whether the distribution of the activity is proportional to the distribution of the SPTLC2 subunit. Fig. 1D demonstrated that SPTLC2 is much more enriched in mitochondria rather than MAM. If the new data indicate the SPT activity is not enriched in mitochondria, the authors should reconsider the model, making the special caution that subcellular distributions of SPT subunit proteins are not necessarily equal to the distribution of SPT that represents the activity to condense serine and palmitoyl CoA to produce 3-ketodihydrosphingosine (KDS). Conventional SPT assay methods are described in, for example, BBA 754, 284 (1983), and J Lipid Res, 50, 1237 (2009).

2. Previous studies and the present study (Fig. 2A) indicated that the SPTLC1 protein is crucial for the stability of the SPTLC2 protein. Nevertheless, Fig 1D suggests that mitochondrial SPTLC2 is stable even without the complex formation of the SPTLC1 subunit. Is this interpretation correct? Or, do the authors interpret that the faint SPTLC1 WB signal in the mitochondria fraction is almost equivalent to the strong SPTLC2 signal at the protein molar level?

3. Fig. 4C: For this model, KDS reductase (KDSR) must be distributed to mitochondria. As the authors discussed, KDSR has so far only been reported on the ER membrane (Kihara & Igarashi, 2004), and the authors have not presented any solid data for the mitochondrial distribution of KDSR. Thus, I consider it is too early to put the KDSR reaction on mitochondria. This pathway should be eliminated in Fig. 4C. At the present, there are several other possibilities. For example, KDS itself, not ceramide, might be toxic to mitochondria. It might also be possible that the flow of KDS to mitochondria could be spatial sequestration of KDS to prevent the formation of toxic ceramide in the ER.

Minor comments

4. Two recent important papers on the 3D structure of the human SPT complex appeared in the same issue of Nat Struct Mol Biol: one is by Wang et al. (vol 28, pp240-248, 2021) and another is by Li et al (vol 28, pp249-257, 2021). Why did the authors cite only the former? The authors should acknowledge the two papers in the introduction section.

5. The authors should briefly discuss whether the model of the authors depicted in Fig. 4C is consistent or inconsistent with the model of 3D SPT complexes (SPTLC1/SPTLC2/ssSPTa/ORMDL3) depicted in the above two Nat Struct Mol Biol papers.

October 26, 2021

Re: Life Science Alliance manuscript #LSA-2021-01278-T

Prof. Eric A. Shoubridge
McGill University
Montreal Neurological Institute
& Dept. of Human Genetics
McGill University
Montreal, 3801 University Street H3A 2B4
Canada

Dear Dr. Shoubridge,

Thank you for submitting your manuscript entitled "Serine Palmitoyltransferase Assembles at ER-Mitochondria Contact Sites" to Life Science Alliance. We invite you to re-submit the manuscript with the following revisions:

- Address Reviewer 1's concerns by discussing alternative scenarios, including that the cis-SPT complex in the ER likely at least co-exists and might also have a function
- Address Reviewer 3's remaining minor concerns

Thank you for this interesting contribution to Life Science Alliance. We are looking forward to receiving your revised manuscript.

Sincerely,

Eric Sawey, PhD
Executive Editor
Life Science Alliance
<http://www.lsa-journal.org>

B. MANUSCRIPT ORGANIZATION AND FORMATTING:

Referee #1:

Please note that, as asked by the editor, in my answer I will take in account also the points raised by reviewer 2.

I do appreciate the improvements and the new set of experiments presented in this revised version, including those in KO cells, that reinforce the function of SPTLC1 and SPTLC2 in regulating mitochondrial sphingolipid synthesis. However, the authors propose that SPTLC1 and SPTLC2 control palmitate-induced mitochondrial fragmentation (via direct sphingolipid synthesis on mitochondria) by interacting "in trans" at ER-mitochondrial contact sites (with SPTLC1 being mainly localized in the ER and SPTLC2 in both ER and mitochondria).

Although this hypothesis is very novel and intriguing, the overall data support both the models where SPTLC1 and SPTLC2 could interact "in cis" and "in trans". Unfortunately, to my opinion the provided experiments are not sufficient to specifically validate the model where SPTLC1 and SPTLC2 interact and function "in trans" at ER-mitochondria junctions nor the functional relevance and existence of the "in trans" interaction in physiological conditions. In other words, a clear evidence that the observed effects upon their knockdown depend on their interaction "in trans" (rather than "in cis") is still missing.

We thank the referee for acknowledging the improvements to our manuscript.

We are glad that the referee now notes that our data support the model where SPTLC1 and SPTLC2 interact both in cis and in trans, as this is the key conclusion of our manuscript and is clearly illustrated in our model. Since the in cis interaction of the SPT-complex in the ER has already been established in the literature, in our manuscript we decided to highlight our novel finding of the in trans assembled SPT complex, but we never claimed that the in cis interaction and complex does not exist.

We have now highlighted more the existence and role of ER-localized in cis complex in the text.

Specific comments:

- The super-resolution of SPTLC2-FLAG show a uniform diffuse distribution across the ER and consequently in the ER near mitochondria (MAM). However, there is no enrichment in ER close to mitochondria, as it would be expected if SPTLC2-FLAG was behaving similarly to the endogenous SPTLC2 (shown in Figure 1D to be enriched in the mitochondrial fraction). Thus, the super-resolution of SPTLC2-FLAG is not particularly supporting the "in trans" model.

In our view, the super-resolution images in Figures 3C and S3C clearly show a punctate localization of SPTLC2-FLAG in foci, and not a uniform diffuse distribution. Some foci are on the ER representing ER-localized SPTLC2 in complex with ER-localized SPTLC1, and some foci are on mitochondria at sites where ER is present, representing mitochondria-localized SPTLC2 in complex with ER-localized SPTLC1. Super-resolution microscopy is not a quantitative method to determine protein enrichment in different compartments, as it only shows a small part of a cell. The native PAGE in Fig 3D was performed from equal amounts of subcellular fractions and can be used to evaluate the enrichment of the SPT complex in the light membrane ER fraction (with in cis SPT complex) and heavy membrane fraction containing mitochondria and mitochondria associated ER-membranes MAMs (with in trans SPT complex).

We would also like to note that the experiments shown here are performed in different conditions, as is common in cell biology where the model organism or cell lines are selected according to what fits best to the experimental requirements. The fractionation in Fig 1D (and native PAGE in Fig 3D) was performed from mouse liver since MAM and mitochondria isolation procedures have been well established in this tissue, and we wanted to demonstrate the in vivo relevance of our findings. The super-resolution microscopy was performed in U2OS-cells in Fig 3C (and also in A431 cells in Fig S3C), where tubular ER and mitochondria can be nicely visualized. The cell line or tissue used in each experiment was clearly mentioned in the manuscript. Cells and tissues cannot be expected to show exactly identical results, since different cell lines and tissues have different gene expression profiles, which could lead to different amounts of ER-mitochondria contact sites, and subsequently to slightly different enrichment of SPTLC2

on mitochondria versus on the ER. Our results from different cell lines and tissues are similar and in line with our conclusions.

- The occurrence of SPTLC1- SPTLC2 "in trans" interaction has been shown using artificially targeted constructs (ER-SPTLC2, exclusively targeted to ER membranes, and OM-SPTLC2, exclusively targeted to mitochondrial membranes). However, these experiments do not prove the existence of a physiological interaction "in trans" between SPTLC1- SPTLC2 at ER-mitochondrial contacts. Also, the native PAGE experiment presented here to support the authors "in trans" model could be interpreted differently: the absence of the complex in the pure mitochondria fraction (and presence of only monomeric SPTLC2) could also be explained by the loss of the main pool of SPTLC1- SPTLC2 interacting "in cis" in the ER, that is not detectable in the purified mitochondria.

As the referee kindly points out, we showed the SPTLC1-SPTLC2 in trans interaction using artificially targeted constructs. The other reviewers did not have any issue with our use of artificially targeted constructs to demonstrate that the two proteins can interact in trans. In the native PAGE experiment, we wanted to study the SPT complex assembly in subcellular fractions (the same fractions which were characterized by SDS-PAGE in Fig 1D) to strengthen our evidence for the in trans and in cis complexes in endogenous physiological conditions. The heavy membrane fraction contains the mitochondria-associated ER membranes (MAMs) and mitochondria, and upon gradient-purification, the heavy membrane fraction is separated into MAMs and mitochondria. As the referee noted in the previous comment and in the first reviews, SPTLC2 is enriched in pure mitochondria (Fig 1D). SPTLC1 is enriched on MAMs (Fig 1D), showing that in the heavy membrane fraction (which consists of mitochondria and MAMs) SPTLC2 and SPTLC1 are on different membranes. In the native PAGE we show a complex of SPTLC1 and SPTLC2 in the heavy membrane fraction, which demonstrates in trans interaction in endogenous physiological condition. We fail to understand how the complex in the heavy membrane fraction could be interpreted as a pool of in cis interacting SPTLC1-SPTLC2, when the referee admits that SPTLC2 is in mitochondria in this fractionation, and the localization of SPTLC1 to MAMs was never questioned. The native PAGE experiment also shows the in cis assembled complex of SPTLC1 and SPTLC2 in the ER-enriched light membrane fraction.

- It is possible that the observed phenotypes in KO cells are due to the loss of SPTLC1- SPTLC2 interacting "in cis" in the MAM at ER-mitochondria contacts. SPTLC1- SPTLC2 interacting "in cis" could form a macromolecular complex with several mitochondria proteins that might contribute to their localization at MAM.

We showed in Fig 3B that ER-targeted SPTLC2-FLAG does not interact with mitochondrial proteins TOMM40 and TOMM20, while WT-SPTLC2-FLAG and mitochondrial outer membrane-targeted SPTLC2 showed these interactions. This result challenges the alternative possibility raised here by the referee.

- Quantitative aspects are still missing (see Reviewer 2 comments): no comparison of the specific activities of SPTLC1- SPTLC2 assembled in cis versus SPTLC1- SPTLC2 assembled in trans, has been shown.

There are several reasons why the activity assays are not feasible to perform and would be inconclusive which we detailed in our response to the first reviews.

Minor:

- It would have been good to show co-localization of SPTLC1- SPTLC2 at ER-mitochondria contacts, even in co-overexpression (given the lack of endogenous antibody working for IF for SPTLC2).

Since we showed the super-resolution images of SPTLC2-FLAG localization at near endogenous protein level, we felt that this was a better approach than overexpression.

- Page 6 line 13: "line" instead of "ine".

Thank you for pointing out the typo. This was corrected on page 7 line 13.

Referee #3:

Comments to EMBOR-2021-52837V2

I found the manuscript has been greatly improved by the revision and I was almost satisfied with the quality of work presented. However, the current manuscript has a few minor concerns, which are listed below. I consider these concerns should be removed before acceptance.

We are grateful to referee #3 for seeing that our manuscript has been strengthened and for supporting the publication.

1) p3, Fig. 1A, and others: Although sphingaine may be a convenient term to represent an isoform of sphingoid bases, dihydrosphingosine is a more systematic nomenclature to represent the sphingoid base isoform, compared with sphinganine (e.g., Hanada, BBA, 2003, 1632, 16-30). Thus, the term of sphingaine should better be changed to dihydrosphingosine throughout the manuscript. KDSR is after 3-keto-dihydrosphingosine reductase, not 3-keto-sphingaine reductase (p4). Of note, dihydroceramide terminologically means N-acylated dihydrosphingosine, not N-acylated sphinganine.

We acknowledge that many synonyms for lipid species are used in the literature, but sphinganine is the correct systematic name as standardized by Lipid Maps (<https://www.lipidmaps.org/databases/lmsd/LMSP01020001?LMID=LMSP01020001>), which is why we did not change it in the text.

2) p13, 1% (4 g/g) digitonin: what does (4 g/g) mean? Do the authors mean 1% (weight/weight)? The manufacturer information should be added because digitonin is a crucial reagent in this study.

Digitonin manufacturer information has been added to the Materials and methods. We have clarified the details on digitonin concentration to 4 g digitonin / g protein (1% w/v) to make it clear for the readers.

3) As per the response of the authors to my major comment #1: Although I agreed with the answer, the brief explanation should be made in the text for audiences. For example, the following sentences by the authors' answer may be added: "As both SPTLC1 and SPTLC2 are required for SPT complex formation and thus SPT activity, the problem with performing activity assays on gradient-purified mitochondria and MAMs is that there will not be any enrichment of activity due to only one half of the enzyme complex being present in each fraction."

We thank the reviewer for the suggestion. However, as it is unusual in a publication to comment on experiments which were not performed, we opt to leave this sentence out. We think that it is clear from our study (particularly Fig. 3D) that both mitochondria and ER are needed for SPT complex assembly and activity at ER-mitochondria contact site. Also, the referee comments and responses will be available for the readers in case they are interested in a thorough explanation.

November 1, 2021

RE: Life Science Alliance Manuscript #LSA-2021-01278-TR

Prof. Eric A. Shoubridge
Montreal Neurological Institute
& Dept. of Human Genetics
McGill University
Montreal, 3801 University Street H3A 2B4
Canada

Dear Dr. Shoubridge,

Thank you for submitting your revised manuscript entitled "Serine Palmitoyltransferase Assembles at ER-Mitochondria Contact Sites". We would be happy to publish your paper in Life Science Alliance pending final revisions necessary to meet our formatting guidelines.

- please add the Twitter handle of your host institute/organization as well as your own or/and one of the authors in our system
- please add callouts for Figure 4D and E to your main manuscript text

A. FINAL FILES:

B. MANUSCRIPT ORGANIZATION AND FORMATTING:

****It is Life Science Alliance policy that if requested, original data images must be made available to the editors. Failure to provide**

original images upon request will result in unavoidable delays in publication. Please ensure that you have access to all original data images prior to final submission.**

The license to publish form must be signed before your manuscript can be sent to production. A link to the electronic license to publish form will be sent to the corresponding author only. Please take a moment to check your funder requirements.

Sincerely,

November 3, 2021

RE: Life Science Alliance Manuscript #LSA-2021-01278-TRR

Prof. Eric A. Shoubridge
McGill University
Montreal Neurological Institute
& Dept. of Human Genetics
McGill University
Montreal, 3801 University Street H3A 2B4
Canada

Dear Dr. Shoubridge,

Thank you for submitting your Research Article entitled "Serine Palmitoyltransferase Assembles at ER-Mitochondria Contact Sites". It is a pleasure to let you know that your manuscript is now accepted for publication in Life Science Alliance. Congratulations on this interesting work.

DISTRIBUTION OF MATERIALS:

Again, congratulations on a very nice paper. I hope you found the review process to be constructive and are pleased with how the manuscript was handled editorially. We look forward to future exciting submissions from your lab.

Sincerely,
